# DNA damage checkpoint activation impairs chromatin homeostasis and promotes mitotic catastrophe during aging

Matthew M Crane[1]*, Adam E Russell[1], Brent J Schafer[1], Ben W Blue[1], Riley Whalen[1], Jared Almazan[1], Mung Gi Hong[1], Bao Nguyen[1], Joslyn E Goings[1], Kenneth L Chen[1,2,3], Ryan Kelly[1], Matt Kaeberlein[1]*

[1]Department of Pathology, University of Washington, Seattle, United States; [2]Department of Genome Sciences, University of Washington, Seattle, United States; [3]Medical Scientist Training Program, University of Washington, Seattle, United States

**Abstract** Genome instability is a hallmark of aging and contributes to age-related disorders such as cancer and Alzheimer's disease. The accumulation of DNA damage during aging has been linked to altered cell cycle dynamics and the failure of cell cycle checkpoints. Here, we use single cell imaging to study the consequences of increased genomic instability during aging in budding yeast and identify striking age-associated genome missegregation events. This breakdown in mitotic fidelity results from the age-related activation of the DNA damage checkpoint and the resulting degradation of histone proteins. Disrupting the ability of cells to degrade histones in response to DNA damage increases replicative lifespan and reduces genomic missegregations. We present several lines of evidence supporting a model of antagonistic pleiotropy in the DNA damage response where histone degradation, and limited histone transcription are beneficial to respond rapidly to damage but reduce lifespan and genomic stability in the long term.
DOI: https://doi.org/10.7554/eLife.50778.001

*For correspondence:
mcrane2@uw.edu (MMC);
kaeber@uw.edu (MK)

## Introduction

Each cell cycle involves a delicate choreography of duplicating genetic material and cellular organelles, with active mechanisms for apportioning them appropriately between mother and daughter cells. Failures of cell cycle regulation can result in severely compromised fitness or cells that respond improperly to environmental cues and emerge as cancerous precursors (*Hanahan and Weinberg, 2011*). In particular, aneuploidy (the gain or loss of partial or whole chromosomes) can be deleterious to fitness (*Beach et al., 2017*; *Sunshine et al., 2016*) and has been implicated in many different types of cancers (*Gordon et al., 2012*) as well as developmental diseases such as Down Syndrome (*Nagaoka et al., 2012*). Recent work has also documented extensive damage and genomic rearrangements that can result following formation of micronuclei or from telomeric crisis (*Maciejowski et al., 2015*; *Zhang et al., 2015*), and identified the ribosomal DNA (rDNA) sequences as particularly vulnerable to genomic damage (*Flach et al., 2014*; *Xu et al., 2017*).

All cells are constantly challenged by DNA damage, both from external environmental sources such as radiation and internal sources such as errors during replication. To cope with DNA damage, cells have robust surveillance mechanisms which arrest the cell cycle and promote repair (*Smith and Rothstein, 2017*). Double-stranded breaks are a particular challenge to cells and multiple breaks will quickly result in cell death if not corrected (*Mehta and Haber, 2014*). The recognition of a double

stranded break by the DNA damage checkpoint (DDC) is consistent across the cell cycle, but there are subtle differences between repair mechanisms depending on the cell cycle stage (*Shaltiel et al., 2015*). Activation of the DDC by a double stranded break causes a host of changes to aid in DNA repair, including increased chromatin mobility as a result of histone degradation (*Dion et al., 2012*; *Hauer et al., 2017*). Extended activation of the DDC, however, is detrimental to cells and has been linked to genomic instability and tetraploidization (*Davoli et al., 2010*; *Davoli and de Lange, 2012*).

Genome instability is a hallmark of aging that occurs in many different species (*López-Otín et al., 2013*). This increase in instability is characterized by elevated rates of DNA mutations, loss of silencing, transposon activation, double-stranded DNA breaks, and changes to telomere maintenance (*Sen et al., 2016*). Furthermore, many progeric diseases which display apparent increases in the rate of aging are characterized by increased genomic damage and a reduced ability to repair DNA (*Burtner and Kennedy, 2010*). Chromosomal instability that results in aneuploidy is also common during aging and could predispose cells to oncogenic transformation (*Naylor and van Deursen, 2016*).

Budding yeast has served as a powerful model for cellular aging by studying how individual cells change during both their replicative and chronological lifespans (*Longo et al., 2012*). Replicative lifespan is defined as the number of daughter cells produced by a mother cell prior to irreversible cell cycle arrest (*Mortimer and Johnston, 1959*). Several types of molecular damage have been associated with replicative aging in yeast mother cells, including mitochondrial dysfunction (*Veatch et al., 2009*), loss of vacuolar pH homeostasis (*Hughes and Gottschling, 2012*), protein oxidation and misfolding (*Hanzén et al., 2016*), and instability at the rDNA (*Ganley et al., 2009*; *Sinclair and Guarente, 1997*).

The rDNA, in particular, is a locus that experiences genome instability during replicative aging due to its makeup of 100–200 tandemly arrayed copies of an identical 9.1 kb repeat. These repeats are prone to recombination events that can lead to the formation of self-replicating, asymmetrically inherited extrachromosomal rDNA circles (ERCs), which accumulate in old mother cells (*Sinclair and Guarente, 1997*). Recombination at the rDNA locus increases dramatically with age and has been strongly associated with loss of Sir2-dependent rDNA silencing and ERC accumulation (*Li et al., 2017*; *Morlot et al., 2019*). Whether ERCs directly cause replicative aging or simply reflect underlying rDNA instability, perhaps caused by transcription of non-coding sequences within the rDNA (*Saka et al., 2013*) or excess rRNA production (*Morlot et al., 2019*), remains a point of inquiry; however, reducing the formation of ERCs and enhancing rDNA stability through deletion of the gene encoding the replication fork block protein Fob1 is sufficient to increase lifespan (*Defossez et al., 1999*). Overexpression of the sirtuin deacetylase Sir2, which also promotes rDNA stability and silences transcription within the rDNA (*Saka et al., 2013*), is similarly sufficient to increase lifespan (*Kaeberlein et al., 1999*), further supporting the model that rDNA instability contributes to replicative aging in yeast.

Beyond rDNA, the loss of genomic stability during yeast replicative aging results in altered cellular function with wide ranging consequences. Changes in silencing (*Kaeberlein et al., 1999*; *McMurray and Gottschling, 2003*) and alterations in nucleosome occupancy (*Hendrickson et al., 2018*; *Hu et al., 2014*) and chromatin remodelling (*Dang et al., 2014*) are thought to underlie some of the large-scale changes in gene expression during aging (*Janssens et al., 2015*). Among these changes in gene expression is a reduction in the levels of key homologous recombination proteins (*Pal et al., 2018*), however mutation accumulation does not appear to be a cause of replicative aging in yeast (*Kaya et al., 2015*). These prior studies have largely focused on cross-sectional, population-level dynamics of yeast aging. By using microfluidic tools to enable single cell, whole lifespan experiments, we have probed genomic instability over the entire lifespan of individual cells. This has revealed striking age-associated genomic instability that can result in the complete loss of genomic content in aging mother cells and is a direct consequence of how the DDC functions to ensure rapid repair of DNA damage.

## Results

### Reversible genome missegregation is common during mother cell aging

In order to begin to understand the impact of aging on cell cycle dynamics and nuclear structure, we measured genome replication and partitioning throughout the mother cell's replicative lifespan by imaging cells expressing fluorescently tagged histone 2B (Htb2:mCherry). To do this, we utilized a microfluidic device which retains mother cells for their entire lifespans while removing daughters via fluid flow (*Crane et al., 2014*). During each cell cycle, the amount of Htb2 in the mother cell nucleus increases during S-phase as histones are transcribed, and then drops as the cell enters mitosis and chromosomes are segregated to the newly formed daughter. The vast majority of cell divisions in young cells follow this characteristic pattern (*Figure 1A*). As cells age, however, abnormal segregation events become common (*Figure 1B*, *Videos 1–4*, please ensure volume is on for all video playback to hear audio explanation). The single cell trace shown in *Figure 1C*, for example, shows a cell undergoing multiple cell cycles with proper division until an abnormal segregation occurs in which the majority of detectable histones are sent to the daughter cell. These genome-level missegregation (GLM) events result in cell cycle arrest that can range from a few minutes (*Figure 1C*-top) to many hours (*Figure 1C*-middle), before they are usually corrected by returning the aberrantly segregated genetic material to the mother cell. If corrected, mother cells are able to proceed through subsequent divisions, but if not, the mother cells will terminally exit the cell cycle and senesce (*Figure 1C*-bottom).

Due to the striking and unexpected nature of the observed age-related GLMs, we wished to confirm that they are not caused by our imaging protocol. These periodic, elongated cell cycles have been long known to occur during replicative lifespan analysis by manual microdissection of yeast cells under a light microscope, often described in the literature as 'symmetric divisions' (*Jazwinski et al., 1989*; *Kennedy et al., 1994*), but the underlying molecular mechanisms have been, until now, completely unknown. In our device, GLM dynamics were not influenced by the fluorophore used or which histone is tagged, as the dynamics of both Htb2:mCherry and histone 2A tagged with GFP (Hta2:GFP) did not differ (*Figure 1—figure supplement 1*). GLM frequency is not an artifact of our imaging protocol, as modifying the fluorescence excitation power or the cumulative excitation energy had no effect on these observations (*Figure 1—figure supplement 2*). GLMs are not caused by the tagging of histones, as imaging strains containing only GFP tagged microtubules showed similar GLM rates and age-related dynamics (*Figure 1—figure supplement 1*). For clarity, the strain containing Htb2:mCherry is referred to as wild-type hereafter. To confirm that the histones do indeed co-localize with DNA during these events, we imaged old mothers and observed the dynamics of Htb2 in cells exposed to the DNA stain Hoechst 3342. As can be clearly seen (*Figure 1—figure supplement 3*, *Video 5*), both the DNA and histones move in concert during these events.

In order to understand the nature of GLM events in detail, we observed aging cells co-expressing the fluorescent histone marker (Htb2:mCherry) with fluorescent markers of the nuclear periphery (Nup49:GFP, *Video 1*), spindle pole bodies (Spc72:GFP, *Video 2*), bud-neck (Myo1:GFP, *Video 3*) or microtubules (Tub1:GFP, *Video 4*). Co-expression of Htb2:mCherry with Nup49:GFP allowed us to observe the nuclear periphery during GLM events, and compare normal divisions with GLMs that are either corrected (*Figure 1D*-top, *Video 1*) or result in terminal GLMs (*Figure 1D*-bottom, *Video 1*). The dynamics of the histone missegregation and recovery can be clearly seen in these time-lapse series, and strikingly the mother cells retain an intact nuclear envelope during these events – even when they appear to lose all of their chromatin (*Figure 1D*). Passage of the histones fully into the daughter cell is evident from cells co-expressing a bud neck marker (Myo1:GFP) along with Htb2:mCherry (*Video 3*). Interestingly, during these events, both spindle poles often fully enter the daughter rather than remain at the bud neck (*Figure 1E*), as can be seen by following the spindle component Spc72 (*Video 2*). Spindle poles frequently move far away from the bud neck (*Figure 1E*). In uncorrected, terminal GLMs, both spindle poles remained in the daughter cell during all observed events (*Figure 1E*, *Video 2*). This can also be observed in videos where tubulin is tagged with GFP (Tub1:GFP), and all of the detectable nuclear microtubules enter the daughter cell during GLMs (*Video 4*).

To quantitatively determine the frequency and penetrance of GLMs during aging, we imaged several hundred mother cells over their entire replicative lifespans, with birth events, GLMs and corrections manually annotated. When cells are young, they have a low probability of experiencing a GLM;

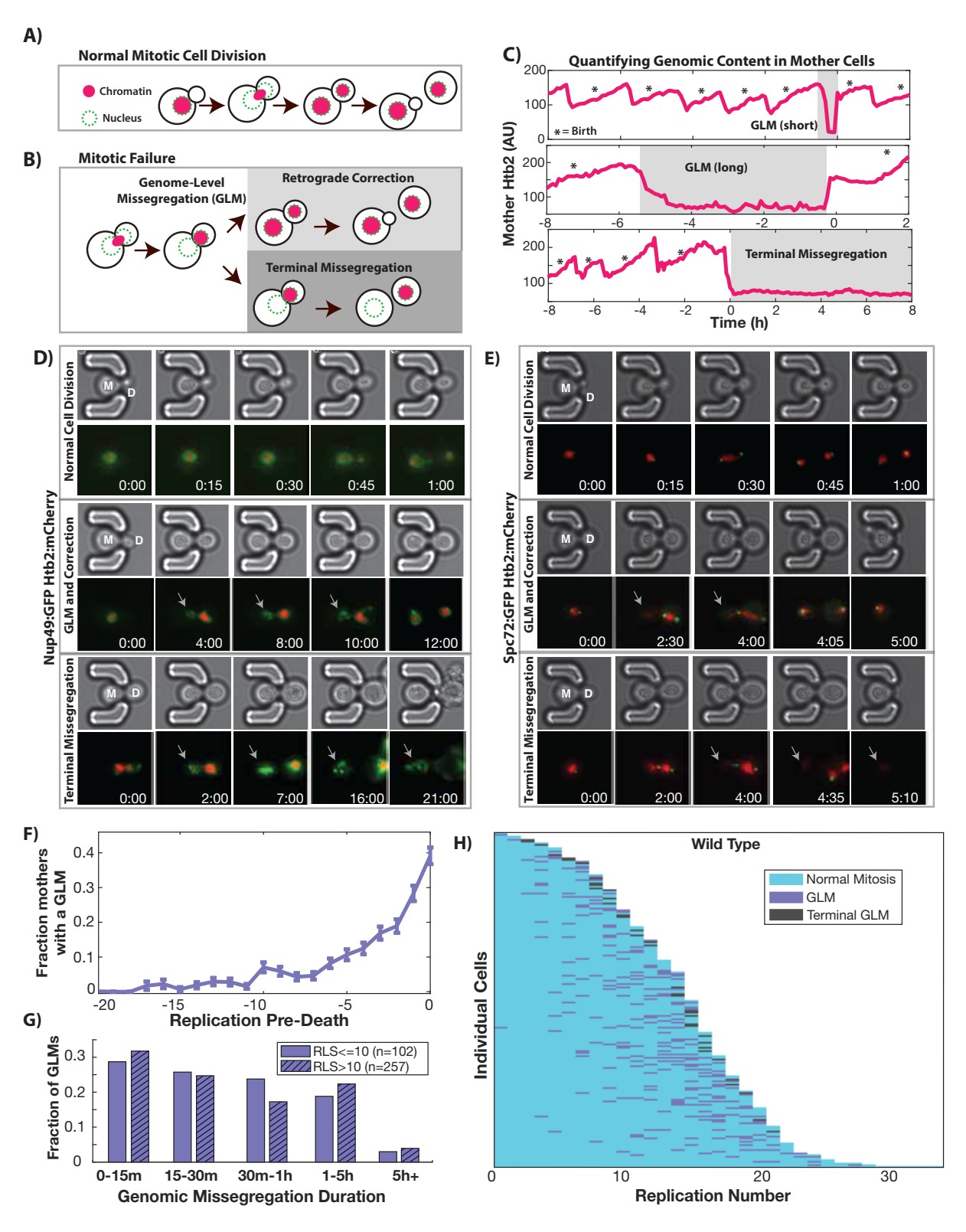

**Figure 1.** During replicative aging cells frequently undergo dramatic genomic missegregation events. (**A**) Schematic showing the process of a normal cell division where chromatin (red) doubles during S-phase and is divided between mother and daughter during mitosis. (**B**) Aging cells frequently experience Genome Level Missegregation (GLM) events where most genomic material enters the daughter while the nuclear envelope appears in both cells. Usually this missegregation is corrected through retrograde transport of genomic material back into the mother cell (top), allowing mother cells to

*Figure 1 continued on next page*

*Figure 1 continued*

go on to divide and produce more daughters. If not corrected and cytokinesis occurs (bottom), this becomes a terminal event wherein mother cells replicatively senesce. (C) Representative single cell traces of mother Htb2 levels showing missegregation (shaded) and active retrograde correction events. GLMs can be resolved quickly (top) or resolution can take hours (middle). A GLM becomes terminal (bottom) if it is not corrected. (*) indicates the formation of new buds, and both cells where the GLM is corrected produce additional daughters. AU indicates arbitrary units. (D) Time-lapse dynamics showing a normal cell division (top, mother cell replicative age 6), a GLM that is corrected (middle, mother cell replicative age 14) and a terminal missegregation (bottom, mother cell replicative age 12) in cells co-expressing Htb2:mCherry and Nup49:GFP. During both GLMs the nuclear envelope is clearly visible in both mother (M) and daughter (D) cells. See *Video 1*. (E) Time-lapse dynamics showing a normal cell division (top, mother cell replicative age 12), a GLM that is corrected (middle, mother cell replicative age 13) and a terminal missegregation (bottom, mother cell replicative age 16) in cells expressing Htb2:mCherry and Spc72:GFP. Both spindle poles can be seen to enter the daughter (D) during these events, and during the correction event a spindle pole returns to the mother (M). In the terminal missegregation, the spindle pole fails to reenter the mother cell. See *Video 2*. Times are indicated in hours:mins from the start of the displayed time-lapse, not the start of the experiment. Arrows indicate mother cells which have lost DNA via a GLM. (F) Missegregation probabilities increase dramatically near the end of replicative lifespan. n = 410 mother cells examined, and error bars are SEM. (G) Many GLMs are corrected within an hour, but some events can last several hours, and the duration of events is not influenced by the replicative age of the mother cell (p>0.05, Student's t-test). Terminal missegregation events were excluded from the analysis. (H) Survival curve showing the dynamics of individual wild-type mother cells. Each row is a separate mother cell, and the color indicates whether a cell experienced a normal cell cycle, GLM or terminal missegregation (n = 200 randomly selected cells).

DOI: https://doi.org/10.7554/eLife.50778.002

The following figure supplements are available for figure 1:

**Figure supplement 1.** GLMs increase at end of life regardless of histone tagged and fluorophore used.
DOI: https://doi.org/10.7554/eLife.50778.003
**Figure supplement 2.** GLMs are not caused by experimental conditions.
DOI: https://doi.org/10.7554/eLife.50778.004
**Figure supplement 3.** DNA moves in concert with histones during GLM events and are correlated with histone levels at the single cell level.
DOI: https://doi.org/10.7554/eLife.50778.005
**Figure supplement 4.** Resolution of GLMs is important to achieve a full lifespan, and GLMs are anti-correlated with remaining lifespan.
DOI: https://doi.org/10.7554/eLife.50778.006
**Figure supplement 5.** G1 duration is not linked to GLM events at the single cell level.
DOI: https://doi.org/10.7554/eLife.50778.007

however, approximately three quarters of mother cells experience one or more GLMs during their replicative lifespan. Furthermore, as cells approach the end of life, the probability of a GLM increases dramatically (*Figure 1F*). The range of arrest durations is broad, with most events resolved within an hour, but some lasting many hours (*Figure 1G*). Interestingly, the duration of each event is not affected by the age of the mother cell (*Figure 1G*). About 90% of GLMs are corrected successfully, allowing individual mother cells to live approximately 30% longer on average than if all GLMs were terminal (*Figure 1—figure supplement 4*). However, even when corrected, mother cells that undergo a GLM are more likely to die in the near future than cells of the same age that have not experienced such an event, and GLMs become increasingly predictive of impending mortality with increasing age (*Figure 1—figure supplement 4*). Similarly, cells that have undergone prior GLM events are more likely to undergo an additional event, indicating that there is a history dependence to GLM events and they do not occur in a completely stochastic manner (*Figure 1—figure supplement 4*). To determine whether changes in cell cycle time, specifically G1 duration, influenced or predicted GLM events, we imaged cells containing HTB2:mCherry and WHI5:GFP. Using the localization of Whi5 to the nucleus as a measure of G1 duration, we confirmed earlier reports that both G1 duration and the fraction of the cell cycle spent in G1 increases during aging. No difference in G1 duration, however, was identified between cell cycles that underwent GLMs and those that did not. The heterogeneity of age-associated mitotic breakdown at the single-cell level can be easily seen when observing the dynamics of all GLM events as a function of age (*Figure 1H*).

## rDNA instability is associated with, but not causal for, the age-related increase in genome missegregation

In order to confirm that individual chromosomes are segregated to daughter cells during GLMs, we directly imaged chromosome positioning in live cells by utilizing a system where TetO repeats are located near the centromere of each chromosome, and TetR:GFP is expressed in the same cell and binds to these repeats allowing visualization of chromosomes throughout the cell

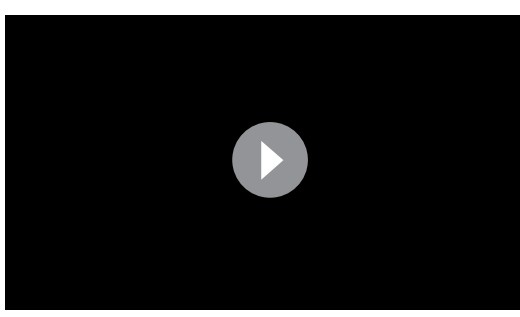

**Video 1.** Normal divisions and GLM dynamics in a strain expressing Htb2:mCherry and Nup49:GFP. **Cell 1:** This cell undergoes six divisions, with histone and nuclear envelope behavior that is characteristic of young, healthy cells. **Cell 2:** Corrected GLM. initial GLM can be seen at timepoint 3 hr:30 min, and the correction at 8 hr:30 min. Following correction, the mother cell is able to bud again at 12 hr, but the nuclear morphology of the daughter (for example, at 16 hr) is significantly altered. **Cell 3:** Terminal GLM. At 3 hr, the mother cell can be seen to undergo a missegregation event. At 15 hr:30 min, the daughter cell buds and can be seen to undergo mitosis, indicating that the daughter cell has separated from the mother. The mother cell eventually dies at 40 hr. The blue arrow points to the mother cell during timepoints where it is experiencing the GLM event. Timestamp is Hours:Min.
DOI: https://doi.org/10.7554/eLife.50778.008

DNA replication, the protein Fob1(fork block 1) binds in the rDNA and prevents collisions between replication forks. Fob1 also promotes recombination independent of fork blocking activity (*Ward et al., 2000*). Deletion of *FOB1* reduces rDNA recombination about 10-fold and significantly extends replicative lifespan (*Defossez et al., 1999*). To explore whether the increase in GLM events during aging was primarily determined by rDNA instability during aging, we removed *FOB1*. These cells experienced a reduction in GLM rates compared with wild-type, but the age-related trend still held true even in *fob1Δ* cells (*Figure 2E,F*). In spite of the increased replicative lifespan, there was no reduction in the fraction of cells that died from a terminal missegregation (*Figure 2—figure supplement 1*), suggesting that rDNA instability is not the dominant cause of GLM events.

The spindle assembly checkpoint delays transition from metaphase to anaphase if chromosomes are not properly attached to the spindle and under tension, and has been shown to delay chromosome condensation (*Kruitwagen et al., 2018*). Because imaging of Chr XII showed that the rDNA remained behind in the mother, while

cycle (*Rohner et al., 2008*). Two different chromosomes were examined, and in each case both copies of Chr IV and Chr V are missegregated to the daughter cells during GLMs (*Figure 2A,B*), as would be expected from the chromatin dynamics (*Figure 2A,B Video 6*). Because Chr XII contains all copies of the rDNA repeats, which are both late replicating and prone to increased instability during aging (*Fangman and Brewer, 1991*; *Sinclair and Guarente, 1997*), we speculated that Chr XII might behave differently from other chromosomes. To assess this, we directly observed Chr XII by targeting a LacI:GFP reporter to LacO sites engineered on the right arm of Chr XII (*Ide et al., 2010*). During GLMs where the majority of DNA enters the daughter cell, both Chr XII chromatids containing the rDNA repeats remain behind in the mother cell (*Figure 2C,D*, *Video 6*). Furthermore, during these GLMs, Chr XII sister chromatids appear as a single point, only separating into two distinct foci following a GLM correction (*Figure 2C*, *Video 6*).

High rates of recombination among tandem repeats of the rDNA make Chr XII particularly susceptible to genomic instability (*Lindstrom et al., 2011*; *Sinclair and Guarente, 1997*), loss-of-heterozygosity (*McMurray and Gottschling, 2003*), and translocations (*Hu et al., 2014*) during aging. To promote unidirectional

**Video 2.** Normal divisions and GLM dynamics in strain expressing Htb2:mCherry and Spc72:GFP. **Cell 1:** Spindle pole dynamics during normal cell divisions. **Cell 2:** A normal healthy division, followed by GLM that is corrected. The two green dots indicate the spindle poles, and at numerous timepoints both poles enter the daughter cell. **Cell 3:** Terminal GLM. The two green dots indicate the spindle poles, and both poles enter the daughter around 2h40m. The poles move around and are highly active, with one at times reentering the mother cell. Finally, at 5 hr:20 m, the daughter cell is washed away indicating it has fully separated from the mother and that this is a terminal GLM. The blue arrow points to the mother cell during timepoints where it is experiencing the GLM. Timestamp is Hours:Min.
DOI: https://doi.org/10.7554/eLife.50778.009

spindle poles and other chromosomes entered the daughter (*Figure 2A–C*), we hypothesized that the GLM arrest could result from improper kinetochore attachment. To test this hypothesis, we deleted the gene encoding the spindle assembly checkpoint component Mad3 (mammalian BubR1). This failed to alter the age-related increase in missegregation, and older *mad3Δ* cells had the same GLM rate as wild type cells (*Figure 2—figure supplement 2*).

## GLMs depend on activation of the metaphase DNA damage checkpoint

Based on the positioning of the spindle poles during GLMs, we hypothesized that cells might be arrested prior to anaphase as a result of the DNA damage checkpoint (DDC). One measure of progression through mitosis is the magnitude of spindle pole separation. We quantified spindle pole separation during GLM events and throughout the entire course of the arrest, spindles remained separated at a consistent 2–5 µm (*Figure 3A*), similar to the mid-anaphase arrest identified by the Bloom lab (*Yang et al., 1997*; *Yeh et al., 1995*). This separation is maintained despite both spindle poles frequently moving far into the daughter bud for at least a portion of the cell cycle (*Figure 1E*, *Figure 3B*).

To further determine whether cells had entered anaphase we monitored localization of Cdc14 (*Figure 3C*), which is localized to the nucleolus for the majority of the cell cycle but exits the nucleolus to initiate anaphase as part of the Cdc-Fourteen Early Anaphase Release (FEAR) and the Mitotic Exit Network (MEN) (*Rock and Amon, 2009*). Cdc14 is specifically required for condensation and segregation of repetitive DNA sequences including the rDNA and telomeres (*D'Amours et al., 2004*; *Sullivan et al., 2004*), and we hypothesized that this role could explain the Chr XII dynamics during GLMs (*Figure 2C*). Furthermore, Cdc14 was recently identified as the limiting step in anaphase, and separately it was observed that compaction of rDNA within the nucleolus interfered with proper release of Cdc14 from the nucleolus (*de Los Santos-Velázquez et al., 2017*; *Roccuzzo et al., 2015*).

In 'normal' cell cycles Cdc14 begins to exit the nucleolus prior to division of genomic material between mother and daughter cells (*Figure 3D*, *Video 7*). In divisions where a cell undergoes a GLM, however, Cdc14 remains localized to the nucleolus during the GLM but is released immediately preceding correction (*Figure 3D*, *Video 7*). The continued localization of Cdc14 to the nucleolus during a GLM event indicates that the FEAR network has yet to

**Video 3.** Normal divisions followed and GLMs in a strain expressing Htb2:mCherry and Myo1:GFP. **Cell1:** The mother cell undergoes four normal divisions, and on the fifth (at timepoint 7 hr:35 min), it experiences a GLM. The bud neck is clearly maintained until the retrograde transport occurs at 12 hr. Following this event, the bud neck is quickly removed, and is completely gone by 12 hr:15 min. **Cell 2:** The bud neck is clearly maintained until the retrograde transport occurs at 2 hr:40 min. Following this event, the bud neck is quickly removed, and is completely gone by 2 hr:55 min. The blue arrow points to the mother cell during timepoints where it is experiencing a GLM. Timestamp is Hours:Min.
DOI: https://doi.org/10.7554/eLife.50778.010

**Video 4.** Normal divisions and GLM dynamics in a strain expressing Htb2:mCherry and Tub1:GFP. **Cell 1:** The mother cell undergoes four divisions normally and on the fifth, at timepoint 7 hr:50 min it experiences a missegregation event that is resolved correctly at 10 hr:30 min. **Cell 2:** Terminal GLM. At timepoint 3 hr:15 min the mother experiences a missegregation event, and both the chromatin and microtubules can be seen entering the daughter cell. At 7 hr:25 min the daughter cell is washed away indicating the cell has completed cytokinesis. The blue arrow points to the mother cell during timepoints where it is experiencing the GLM. Timestamp is Hours:Min.
DOI: https://doi.org/10.7554/eLife.50778.011

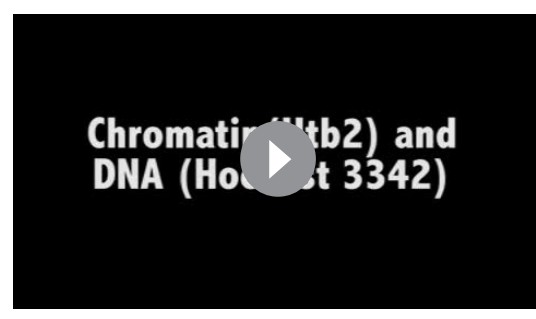

**Video 5.** DNA co-localizes with tagged histones through mitosis and during a GLM. Cells expressing Htb2:mCherry were stained with Hoechst 3342, a live DNA stain. The first part of the video shows an overlay of red (Htb2:mCherry) and blue (Hoechst 3342), and the second shows the channels separated. Cell 1: In a normal cell cycle, the histones co-localize with the DNA, and both increase or decrease in fluorescence in the mother cell simultaneously. Cell 2: During a GLM the histones co-localize with the DNA, and both increase or decrease in fluorescence in the mother cell simultaneously.

DOI: https://doi.org/10.7554/eLife.50778.012

initiate anaphase, and Cdc14 activation by FEAR precedes the return of genomic material to the mother. By pooling and averaging cell cycles where mitosis occurs normally, the mother cell histone content can be seen to fall from 2N to 1N as Cdc14 exits the nucleolus following initiation of anaphase (*Figure 3E*). In events, where a GLM occurs, however, Cdc14 remains in the nucleolus even while the majority of the chromatin is in the daughter (*Figure 3F*, *Figure 4—figure supplement 1*). Only following Cdc14 release from the nucleolus does the chromatin exit from the daughter cell and return the mother to 1N (*Figure 3D*). Because Chr XII requires Cdc14 activity for condensation, this likely explains our results where Chr XII remains behind during GLM events (*Figure 2C*). That the nucleolus (*Figure 3D*) and rDNA (*Figure 2C*) remain in the mother during these events distinguishes them from prior nuclear extensions where the nucleolus acted as a sink and entered the daughter while the chromatin remained behind (*Witkin et al., 2012*). The observation that cells experiencing a GLM arrest prior to anaphase agrees with prior work showing that Cdc14 release during anaphase generates pulling forces within the mother to counteract those in the daughter (*Ross and Cohen-Fix, 2004*).

The observations described above led us to hypothesize that GLM events were associated with activation of the metaphase DNA damage checkpoint. The yeast protein Rad9 (similar to mammalian 53BP1) is a critical component of the cellular response to DNA damage (*Toh and Lowndes, 2003*). Upon DNA damage, Rad9 is hyper-phosphorylated by Mec1 and Tel1 which results in activation of both Chk1 and Rad53 (*Emili, 1998*; *Vialard et al., 1998*). Checkpoint compromised *rad9Δ* cells experience essentially a complete abolition of terminal GLM events (*Figure 3G*). Compared with wildtype cells, *rad9Δ* cells also experience a dramatic reduction in age-associated GLM rates (*Figure 3H*), which can also be seen at the single cell level (*Figure 3I* as compared to *Figure 1H*). To further confirm the causal connection between activation of the DNA damage checkpoint and GLMs, we chemically induced DNA damage in a population of young cells using 500 µg/ml of zeocin. This concentration was previously shown to result in significant activation of the DNA damage checkpoint, and degradation of the histone pool (*Hauer et al., 2017*). Following administration of zeocin, young cells experienced a dramatic increase in GLMs compared with control cells that are not exposed to zeocin (*Figure 3—figure supplement 1*). This demonstrates that GLM events are caused by activation of the DNA damage checkpoint, and that the age-related increase in events is likely due to increases in genomic instability causing increased activation of the DNA damage checkpoint.

## Homologous recombination suppresses age-associated genome missegregation

In order to repair double stranded breaks, the DDC relies upon homologous recombination and non-homologous end joining. In cells that have compromised DNA repair, the DDC is activated for an extended amount of time before cells are able to successfully complete DNA repair and continue through the cell cycle. Because GLMs occur following activation of the DDC, we hypothesized that compromising the ability of cells to perform homologous recombination could increase both the rate and duration of GLMs. To do this we deleted *RAD52* which performs functions analogous to mammalian BRCA2 in homologous recombination. In these *rad52Δ* cells, there is no age-related increase in GLM events, but instead a constant high-probability (*Figure 4A*). This can also be seen at the single cell level (*Figure 4B*). Recent work has shown that there is a loss of homologous repair

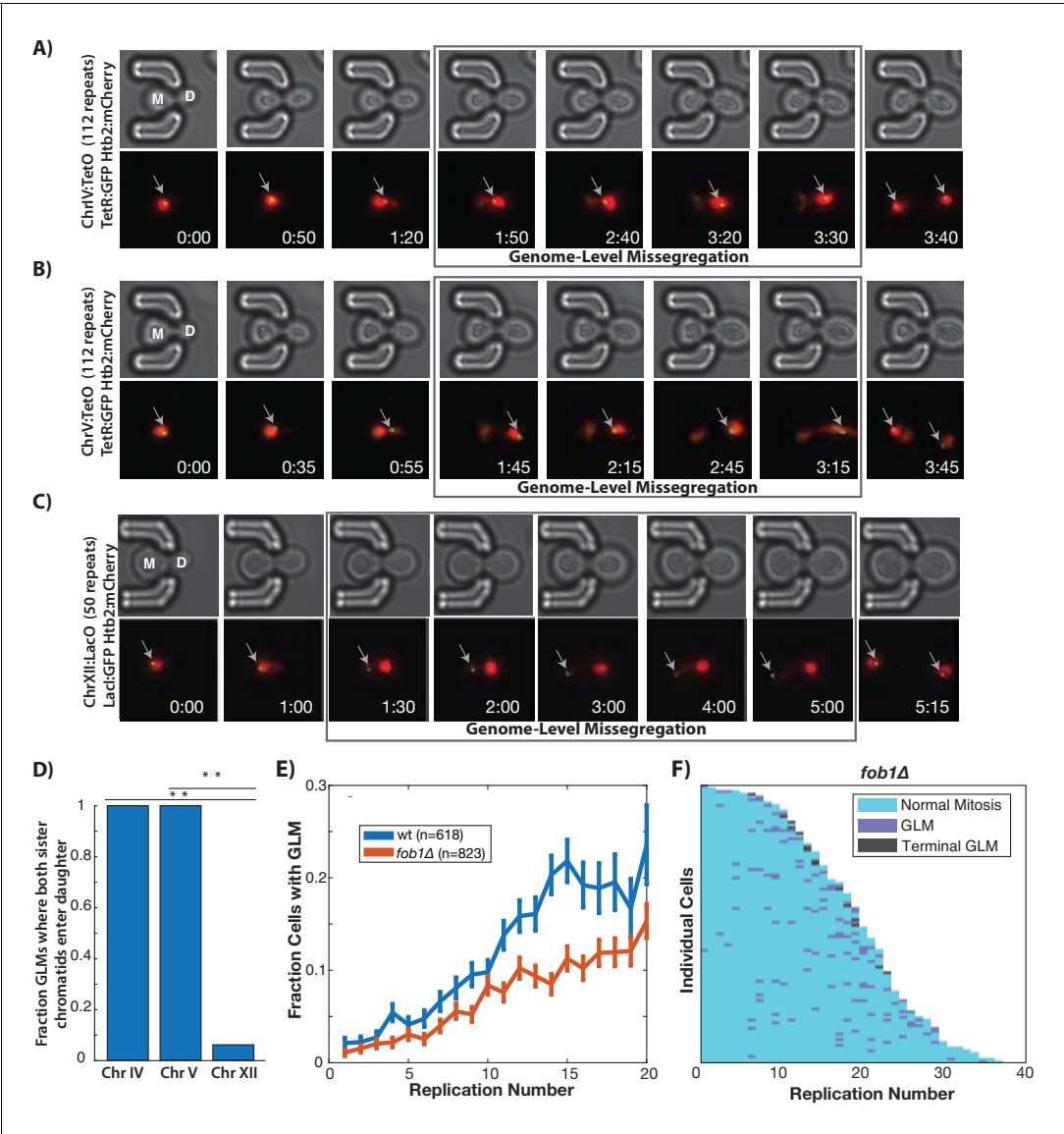

**Figure 2.** GLMs are linked to rDNA instability. (**A**) Direct observation of Chr IV using TetR:GFP and TetO repeats on Chr IV. When the cell experiences a GLM, both chromatids of Chr IV move to the daughter along with the majority of the chromatin. Following correction and anaphase, a single green dot can be seen in both mother (M) and daughter (D) cells. Mother cell replicative age equals five at the beginning of this timelapse. GFP contains an NLS to increase the nuclear concentration, and is only localized to a dot at the site of the chromosome. (**B**) Direct observation of Chr V using TetR:GFP and TetO repeats on Chr V. When the cell experiences a GLM, both chromatids of Chr V move to the daughter along with the majority of the chromatin. Following the correction, a single green dot can be seen in both mother and daughter cells. Mother cell replicative age equals 19 at the beginning of this timelapse. (**C**) Direct observation of Chr XII using LacI:GFP and LacO repeats on Chr XII. Unlike the other chromosomes, when the cell experiences a missegregation event, both chromatids of Chr XII remain behind in the mother. Following correction, a single green dot can be seen in both mother and daughter cells. Mother cell replicative age equals eight at the beginning of this timelapse. The gray arrows mark the location of the labeled chromosomes. Times are indicated in hours:mins. (**D**) Quantification of the fraction of observed GLM events where each chromosome pair entered the daughter or remained in the mother (p<0.001 using bootstrapping with replacement). (**E**) Removal of *FOB1* increases stability at the rDNA and reduces GLM rates, but fails to abolish an increase in GLMs during aging (curve shows mean and error bars are SEM, p<0.05 determined by Cochran Q-test). (**F**) Survival curve showing the GLM dynamics in individual *fob1Δ* mother cells. Each row is a separate mother cell, and the color indicates whether a cell experienced a normal cell cycle, GLM or terminal missegregation (n = 100 randomly selected cells).

DOI: https://doi.org/10.7554/eLife.50778.013

The following figure supplements are available for figure 2:

**Figure supplement 1.** Stabilizing the rDNA by removing FOB1 doesn't reduce the fraction of cells that experience terminal GLMs.

DOI: https://doi.org/10.7554/eLife.50778.014

**Figure supplement 2.** Removing MAD3 (mammalian BubR1) fails to eliminate the age-related increase in missegregation rate.

*Figure 2 continued on next page*

*Figure 2 continued*

DOI: https://doi.org/10.7554/eLife.50778.015

proteins during aging (*Pal et al., 2018*), and this could explain the age-related increase in GLM frequency in wildtype cells (*Figure 4A*). Although *rad52Δ* cells are more likely to suffer a terminal GLM than wildtype cells, this is not due to an increased rate of correction failure (*Figure 4C*). Rather, this is because the cells are so likely to experience many events that eventually one becomes terminal. Also intriguing is that the GLM duration in *rad52Δ* cells is significantly longer than in wildtype (*Figure 4D*). Thus, homologous recombination is necessary to reduce both the frequency and duration of GLMs, but Rad52-dependent homologous recombination does not affect the proportion of GLMs that can be repaired.

## DNA damage-induced histone degradation contributes to GLMs

Histone levels have been shown to influence replicative lifespan in yeast (*Feser et al., 2010*; *Yu et al., 2019*), and recently activation of the DDC was found to cause a dramatic reduction in global histone levels (*Hauer et al., 2017*). We hypothesized that degradation of histones as a result of activation of the DDC might result in age-related genomic instability and GLMs (*Hu et al., 2014*). To test this, we limited the ability of cells to ubiquitinate and degrade histone proteins by separately deleting *TOM1* and *IES4. TOM1* encodes a factor required for degradation of excess histones (*Singh et al., 2009*), and *IES4* is a member of the INO80 chromatin remodeling complex that is necessary for the DDC dependent reduction of histones (*Hauer et al., 2017*). In cells lacking either of these genes, there is a significant reduction in GLM rates at the population level during

aging (*Figure 5A*). Not only do these mutations result in reduced GLM rates, but fewer cells die from terminal missegregations (*Figure 5B*), and this is not a result of an increased fraction of GLM events that are corrected (*Figure 5C*). Unlike mutations that compromise the DDC, however, deletions of *TOM1* or *IES4* do not come at a cost to replicative lifespan, but actually result in an increased lifespan (*Figure 5D,E*).

Histone transcription is tightly regulated and confined to S-phase (*Kurat et al., 2014b*). To directly test the mechanistic link between the histone pool and GLMs, we removed the temporal cell cycle regulation of histone transcription by deleting *HPC2*, which encodes a component of the HIR complex that represses histone transcription outside of S-phase (*Green et al., 2005*). Although deletion of *HPC2* increases replicative lifespan and alters the dynamics of histone transcription, *hpc2Δ* cells do not have higher levels of histone proteins (*Feser et al., 2010*). Deletion of *HPC2* results in a significant reduction in terminal missegregation events (*Figure 6A*), and also a reduced frequency of GLMs in aging cells (*Figure 6B*). This reduction in GLM rates can be clearly seen at the single cell level (*Figure 6C*). To perform a complementary experiment and reduce histone transcription, we deleted *SPT21*, which encodes a protein that positively regulates expression of HTA2-HTB2 (*Dollard et al., 1994*; *Kurat et al., 2014a*). Deletion of *SPT21* has been previously

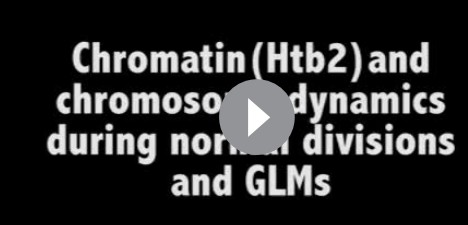

**Video 6.** Chromosome dynamics during GLMs. **Cell 1:** ChrIV dynamics during a GLM. Cell is expressing Htb2:mCherry and TetR:GFP, and has TetO repeats inserted into ChrIV. The Chromosome IV sister chromatids clearly move into the daughter with the majority of the genome. **Cell 2:** ChrV dynamics during a GLM. Cell is expressing Htb2:mCherry and TetR:GFP, and has TetO repeats inserted into ChrV. The Chromosome V sister chromatids clearly move into the daughter with the majority of the genome. **Cells 3 and 4:** ChrXII dynamics during a GLM. Cell is expressing Htb2:mCherry and LacI:GFP, and has LacO repeats inserted into ChrXII. The Chromosome XII sister chromatids clearly remain behind despite the majority of the genome entering the daughter cell. For all movies and cells, sister chromatids separate simultaneous with anaphase initiation and correction of the GLM. Sister chromatids can then be identified in both mother and daughter cells. The blue arrow points to the mother cell during timepoints where it is experiencing the GLM. Timestamp is Hours:Min.

DOI: https://doi.org/10.7554/eLife.50778.016

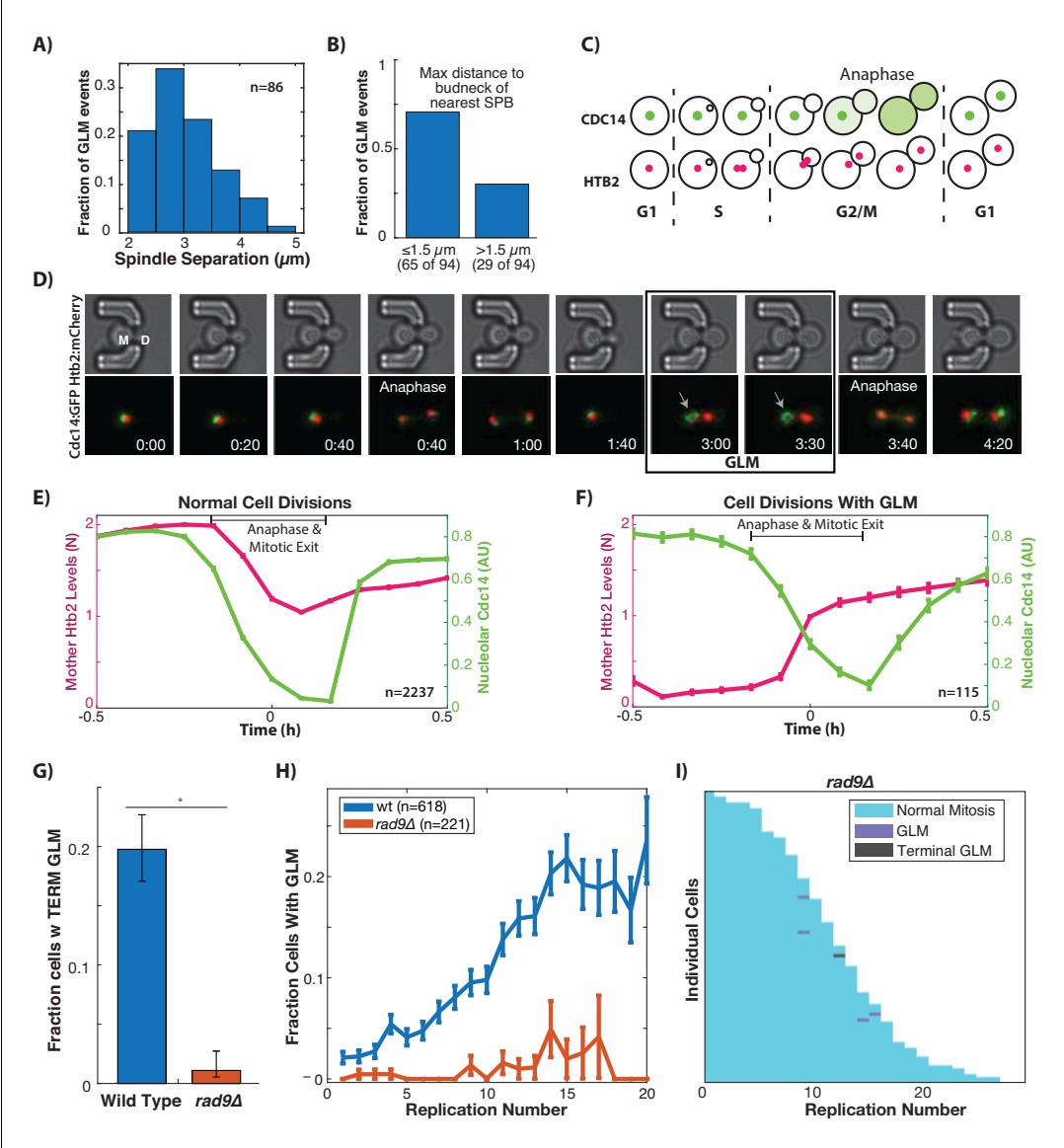

**Figure 3.** GLMs occur during arrest at the metaphase DNA checkpoint. (**A**) Histogram of spindle pole separation for cells arrested during GLMs. Separation was measured at three timepoints for each cell, and the average for each cell was used for the histogram (n = 86). (**B**) Distance from nearest spindle pole body (SPB) to the bud neck during the GLM. Both SPB are in the daughter during these events. (**C**) Schematic showing Cdc14 dynamics during the cell cycle. Cdc14 is localized to the nucleolus, except during anaphase. (**D**) Timelapse images of a single cell containing Cdc14: GFP and Htb2:mCherry that goes through a normal division and then a division with a GLM. The arrow points to nucleolar localized Cdc14 during the GLM that is released to allow anaphase entry. See *Video 7*. Time is hours:minutes. (**E**) Plot shows average of single cell traces of cells undergoing normal cell divisions where both Cdc14 and Htb2 were imaged. Htb2 levels were normalized at the single cell level, so 1 is 1N, and cells cycles were aligned using Cdc14 dynamics. (**F**) Quantification of nucleolar Cdc14 in populations of cells confirms that anaphase entry is delayed in cells experiencing a GLM. Individual cells were aligned to time of correction event. Only GLMs that lasted >30 min were used, and error bars are standard error. (**G**) Mutants with compromised DNA damage checkpoint (*rad9Δ*) have significantly reduced rates of terminal missegregations (**H**) Mutants with compromised DNA damage checkpoint (*rad9Δ*) have no age-related increase in the GLM rate (p>0.05 determined by Cochran Q-test, plot shows mean and error bars are SEM). (**I**) Survival curve showing the significant reduction in GLM rates in individual *rad9Δ* mother cells. Each row is a separate mother cell, and the color indicates whether a cell experienced a normal cell cycle, GLM or terminal missegregation (n = 100 randomly selected cells).
DOI: https://doi.org/10.7554/eLife.50778.017

The following figure supplement is available for figure 3:

**Figure supplement 1.** DNA damage in young cells results in GLMs.
DOI: https://doi.org/10.7554/eLife.50778.018

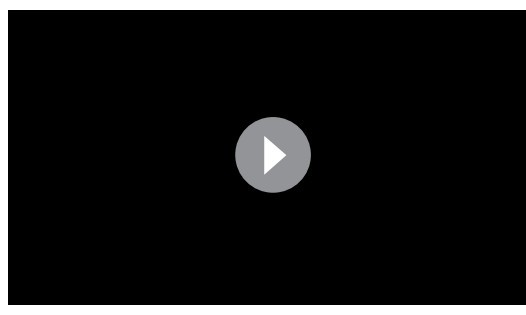

**Video 7.** Normal divisions and GLM dynamics in a strain expressing Htb2:mCherry and Cdc14:GFP. **Cell 1:** Cdc14 dynamics during normal cell divisions. **Cell 2:** Cdc14 remains remains localized to the nucleolus even when the cell experiences a GLM. The exit of Cdc14: GFP from the nucleolus at 2 hr coincides with the correction. The blue arrow points to the mother cell during timepoints where it is experiencing the GLM. Timestamp is Hours:Min.

DOI: https://doi.org/10.7554/eLife.50778.019

shown to reduce histone transcription and increase rDNA instability (*Eriksson et al., 2012*; *Kobayashi and Sasaki, 2017*). In contrast to deletion of *HPC2*, *TOM1*, and *IES4*, deletion of *SPT21* caused an increase in terminal GLMs (*Figure 6A*) and an increased frequency of GLMs in mother cells (*Figure 6B,D*). These observations demonstrate that altering the temporal dynamics of histone transcription in aging cells is sufficient to modulate the frequency of GLM events both upward and downward.

## GLM correction requires the spindle positioning checkpoint but not the spindle assembly checkpoint

Proper segregation of chromosomes during the cell cycle relies on checkpoints to ensure spindle attachment and positioning prior to the initiation of anaphase (*London and Biggins, 2014*). The spindle positioning checkpoint works to ensure that the spindle is properly positioned within the budneck between mother and daughter prior to anaphase (*Fraschini et al., 2008*), and failure of this checkpoint can result in a daughter cell receiving no genomic content (*London and Biggins, 2014*). Given the dynamic movement of the spindle pole bodies into the daughter cell during GLM events (*Figure 1E*), we hypothesized that the spindle positioning checkpoint could be playing a role in the ability of cells to recover from GLM events. To test this, we deleted *BFA1* which is a critical component of the checkpoint (*D'Aquino et al., 2005*; *Geymonat et al., 2003*; *Hu et al., 2001*) and is potentially activated by DNA damage (*Campbell et al., 2019*). This checkpoint has been primarily studied from the context of spindle misalignment within the mother cell. While deletion of *BFA1* did not impact the frequency of GLM events during aging (*Figure 7A*), it significantly increased the fraction of GLMs that were not corrected and resulted in terminal missegregation (*Figure 7B*). The high rate of failures can be clearly observed at the single cell level, where over half of *bfa1Δ* cells die from a terminal missegregation (*Figure 7C*). As discussed earlier, disruption of the spindle assembly checkpoint by deletion of *MAD3* failed to affect the age-related increase in GLM rates (*Figure 2—figure supplement 2*). Thus, neither activation of the spindle assembly nor the spindle positioning checkpoints result in GLMs, but the spindle positioning checkpoint is critical for healthy resolution of GLMs. When combined with our observations on the role of the DNA damage checkpoint, histone degradation and homologous recombination, these findings support a model whereby activation of the DDC can result in a reduced histone pool and a potentially catastrophic loss of genomic material from mother cells that can be corrected by the spindle positioning checkpoint (*Figure 7D*).

## Discussion

Because function declines in many different and subtle ways during aging, catastrophic failures and homeostatic systems like those uncovered here may sometimes only be observed in aged organisms. Imaging of individual yeast cells through microfluidic trapping allowed us to observe GLMs that occur in most mother cells one or more times during their lives. These events are rare in young cells, are caused by activation of the DDC and the resulting histone degradation during metaphase, and are usually successfully resolved by retrograde transport of genomic material back to the mother cell through activation of the spindle positioning checkpoint. These observations demonstrate a dynamic and intricate set of checks and balances that act to maintain genomic integrity during cellular aging.

By providing insights into the dynamics of genomic instability during aging at the single cell level, the data presented here builds on and integrates a wealth of prior observations related to yeast aging which have generally been measured only at the population level. Genome instability, and

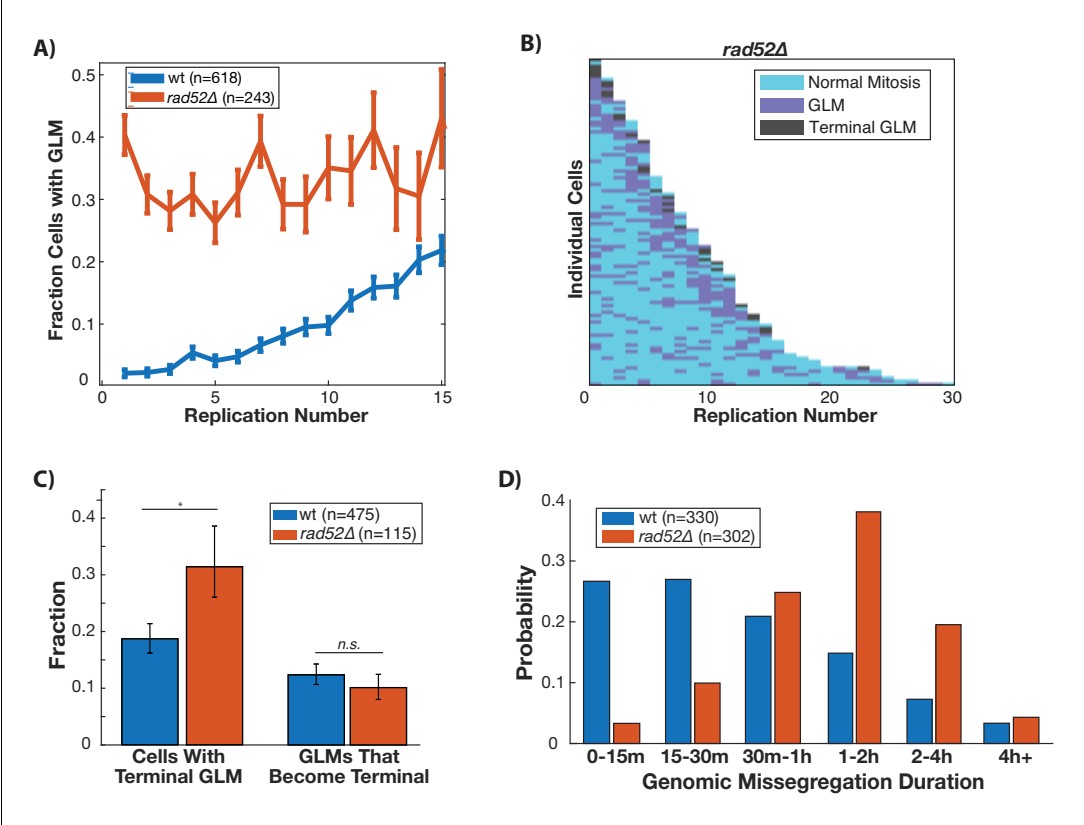

**Figure 4.** Disruption of homologous repair by deletion of RAD52 causes constant high rate of GLMs during aging. (**A**) *Rad52Δ* mutants have a constant high rate of GLMs but do not show a significant age-related increase in GLMs (p>0.05 cochran q test). Lifespan only shown 0–15 divisions due to reduced lifespan of *rad52Δ* mutants. (**B**) Survival curve showing the significant increase in GLM rates in individual *rad52Δ* mother cells. Each row is a separate mother cell, and the color indicates whether a cell experienced a normal cell cycle, GLM or terminal missegregation (n = 100 randomly selected cells). (**C**) *Rad52Δ* cells are more likely than wild-type to experience a terminal GLM, but any individual GLM is not more likely to result a terminal missegregation. This suggests Rad52 is important for preventing GLMs, but not for ensuring genomic content is properly segregated. (**D**) Cells lacking RAD52 have a statistically significant increase in the length of time a GLM lasts relative to wild-type cells (n is the number of cell cycles quantified, p<0.01 Students t-test).

DOI: https://doi.org/10.7554/eLife.50778.020

The following figure supplement is available for figure 4:

**Figure supplement 1.** Cdc14 single cell traces showing terminal and corrected GLM events.

DOI: https://doi.org/10.7554/eLife.50778.021

rDNA instability in particular, has long been thought to be a major contributor to replicative aging in yeast. However, our data indicate that enhancing rDNA stability through deletion of *FOB1* only delays, but does not substantially alter, the prevalence of GLMs or the capacity for cells to appropriately resolve GLMs. This may reflect general genome instability arising from cascading failures during aging (*Gottschling and Nyström, 2017*). Loss of vacuolar pH has been identified as an early in life change that can result in a loss of mitochondrial membrane potential (*Hughes and Gottschling, 2012*), which in turn has been linked to increased genomic instability during aging, likely through altered iron-sulfur cluster production and the resulting DNA replication stress (*Veatch et al., 2009*). This increased replication stress, when coupled with a loss of homologous recombination proteins with age (*Pal et al., 2018*), could result in the increased activation of the DDC we observe. Our data also illustrate how the previously observed decline in histone abundance during aging (*Feser et al., 2010*) likely results from DDC activation in aged mother cells, and that this protective response, which is beneficial in young cells upon exposure to DNA damaging agents (*Gunjan and Verreault, 2003*; *Hauer et al., 2017*; *Liang et al., 2012*), leads to elevated rates of genome missegregation in old cells which eventually causes terminal senescence and death.

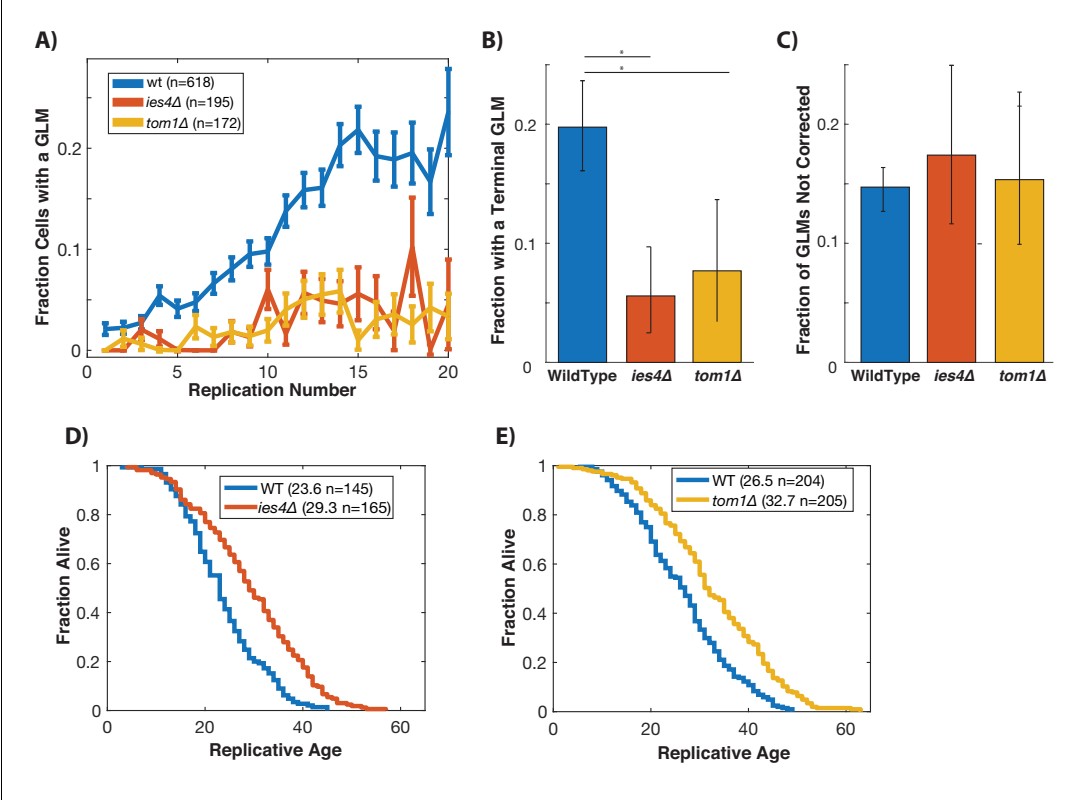

**Figure 5.** Preventing damage-induced histone degradation reduces age-associated GLMs. (**A**) Cells with reduced ability to degrade histone proteins (*ies4Δ* and *tom1Δ*) but do not show a significant age-related increase in GLMs (curve shows mean and error bars are SEM, p>0.05 both strains determined by Cochran Q-test). (**B**) The reduction in GLM rates also results in a reduction in the total number of cells that undergo a terminal missegregation. Error bars are confidence intervals generated by bootstrapping with replacement generated so that non-overlapping bars indicate confidence at the p=0.05 threshold. (**C**) The fraction of GLM events that are not corrected is unchanged in *ies4Δ* and *tom1Δ* cells. Error bars are confidence intervals generated by bootstrapping with replacement generated so that non-overlapping bars indicate confidence at the p=0.05 threshold. (**D**) Eliminating *IES4* results in an increased replicative lifespan (p<0.001 log-rank). Legend shows mean RLS and number of cells measured by microdissection. E)As has been previously reported, deleting *TOM1* results in an increased replicative lifespan (p<0.001 log-rank). Legend shows mean RLS and number of cells measured by microdissection.

DOI: https://doi.org/10.7554/eLife.50778.022

Aging has been associated with an increased cell cycle time, and in particular an increased G1 duration. In stem cells, elongated G1 is linked to increased DNA replication stress (*Flach et al., 2014*). In yeast it is thought to result from high levels of ERCs resulting in increased levels of the Rb analog Whi5 (*Neurohr et al., 2018*). Intriguingly, reducing the G1 duration in yeast by overexpressing Cln2 fails to increase replicative lifespan (*Neurohr et al., 2018*), but results in an improvement in single-strand annealing rates (*Young et al., 2019*). An improvement in DNA repair efficiency might be assumed to result in a reduced frequency of GLM events. Surprisingly, however, at the single cell level we failed to find a connection between G1 duration and the occurrence of GLMs (*Figure 1—figure supplement 5*). The G1 elongation that is a conserved hallmark of aging may serve a purpose in aged cells that has yet to be identified. For example, a short G1 phase imposes constitutive replication stress in cycling stem cells (*Ahuja et al., 2016*). Similarly, a lengthened G1 in aging yeast could lead to improved loading of replication machinery and the improved origin firing duration may compensate for the reduced single-strand annealing.

The striking movements of the spindle pole bodies into the daughter cell during GLMs and then back to the mother cell during resolution further supports the idea that these events represent potentially catastrophic mitotic failures. The metaphase DNA damage checkpoint has been previously implicated in highly dynamic movements of the spindle poles prior to anaphase initiation (*Palmer et al., 1989*; *Yang et al., 1997*; *Yeh et al., 2000*; *Yeh et al., 1995*). In cells that

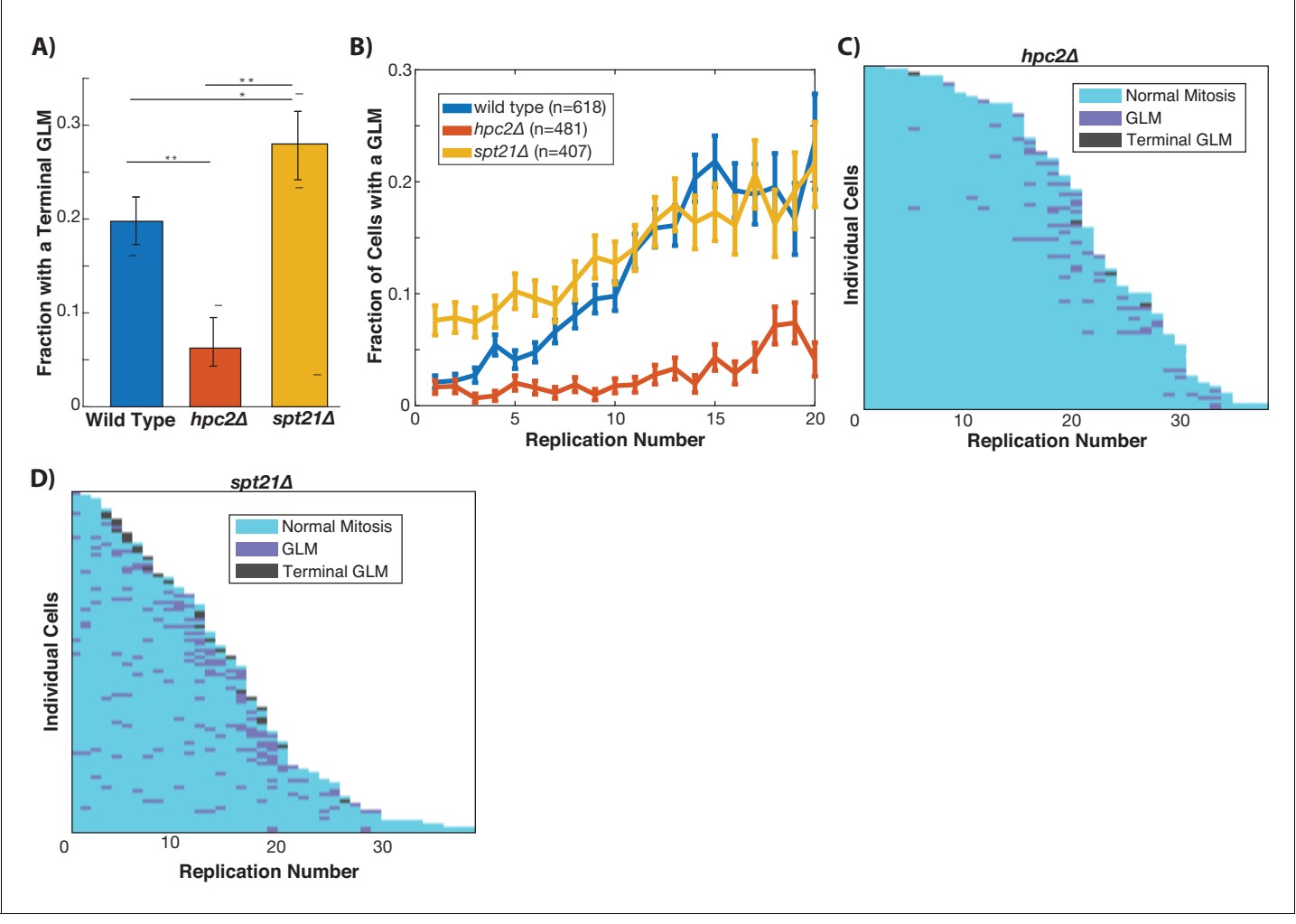

**Figure 6.** Histone transcription influences GLMs rates. (A) Constitutive histone transcription (*hpc2Δ*) significantly reduces GLM rates at the population level when compared with wild-type (p<0.001, **), while cells with reduced ability to transcribe histones (*spt21Δ*) experience increased rates (p<0.05, *) compared with wild type cells. Error bars are confidence intervals generated by bootstrapping with replacement generated so that non-overlapping bars indicate confidence at the p=0.05 threshold. (B) At a population level, manipulating histone transcription affects GLM rates and cells with increased histone transcription (*hpc2Δ*) do not show a significant age-related increase in GLMs (curve shows mean and error bars are SEM, p>0.05 for *hpc2Δ* determined by Cochran Q-test). Error bars are standard error. (C) Survival curve showing the significant reduction in GLM rates in individual *hpc2Δ* mother cells. Each row is a separate mother cell, and the color indicates whether a cell experienced a normal cell cycle, GLM or terminal missegregation (n = 100 randomly selected cells). (D) Survival curve showing the significant increase in GLM rates in individual *hpc2Δ* mother cells (n = 100 randomly selected).

DOI: https://doi.org/10.7554/eLife.50778.023

experience a DNA double strand break, oscillations of the spindle poles and entry into the daughter have been previously reported, but only in the context of mutants with compromised DNA damage checkpoints such as *chk1Δ* or *rad53Δ* (*Dotiwala et al., 2007*). Interestingly, a recent report described segregation of the nucleus and spindle poles into the daughter cell in five aging yeast cells, which is likely to be the same phenotype detailed here (*Neurohr et al., 2018*). Our single-cell data indicate that these events are surprisingly frequent in aging mother cells and the spindle positioning checkpoint is engaged to allow for proper resolution in most cases.

Our findings suggest that responses to DNA damage have evolved under constraints of antagonistic pleiotropy, which refers to genes that have a beneficial effect during youth but whose activity results in detrimental effects later in life (*Williams, 1957*). In particular, degradation of histones by the DDC appears to be a classic example of this. Young (log phase) cells that are unable to degrade histones following DDC activation display an impaired ability to rapidly repair DNA damage and

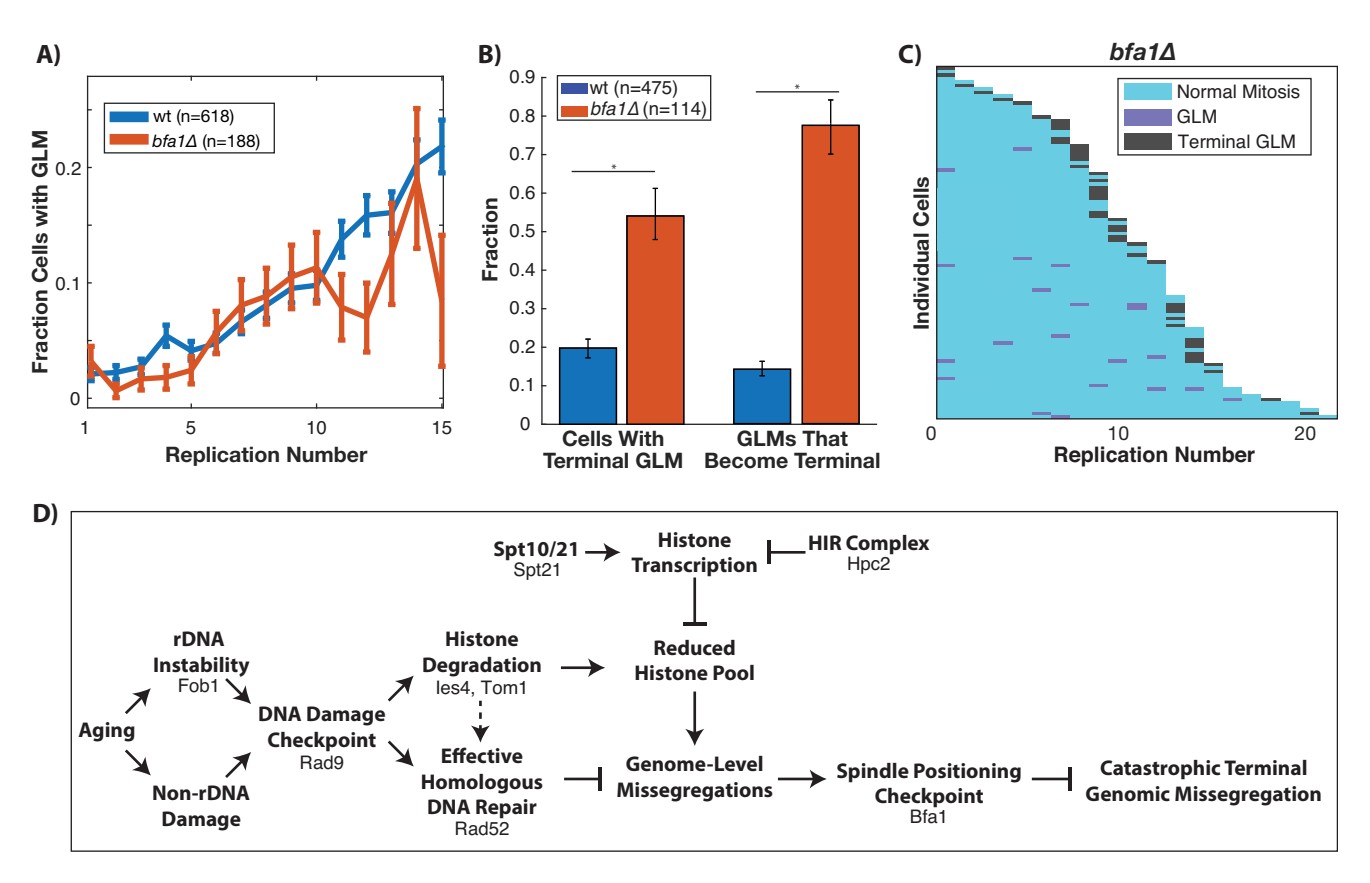

**Figure 7.** Correction of GLM events requires the spindle positioning checkpoint. (**A**) Removal of *BFA1* disrupts the spindle positioning checkpoint, but fails to abolish an increase in GLMs during aging (curve shows mean and error bars are SEM, p<0.05 determined by Cochran Q-test). Lifespan only shown 0–15 divisions due to reduced lifespan of *bfa1Δ* mutants. (**B**) Compared with wildtype, a *bfa1Δ* strain has significant increases in both the fraction of cells that experience terminal missegregations and the likelihood that an individual GLM will become terminal. Error bars that do not overlap show significance at the p=0.05 level and were generated by bootstrapping with replacement. N values are number of cells. (**C**) The difference in *bfa1Δ* GLM dynamics is stark at the single cell level where most GLMs result in terminal missegregations and most mother cells die because of terminal GLMs. Each row is a separate mother cell, and the color indicates whether a cell experienced a normal cell cycle, GLM or terminal missegregation (n = 100 randomly selected cells). (**D**) Model for age-associated GLMs that links age associated changes in DNA damage to failures during mitosis. Proteins included in the diagram at each stage are those that have been genetically perturbed in this work.

DOI: https://doi.org/10.7554/eLife.50778.024

resume cell division following response to exogenous DNA damage (*Gunjan and Verreault, 2003*; *Hauer et al., 2017*; *Liang et al., 2012*). Thus, DDC-mediated histone degradation is likely to provide a selective benefit in young cells which may experience DNA damage due to environmental exposures, for example. However, activation of this response due to age-associated genomic instability becomes detrimental, and preventing histone degradation in response to DDC activation in aged cells both reduces GLMs and increases lifespan. This tradeoff explains the difference in lifespan that can be observed by affecting GLM rates in different ways. Reducing the ability of cells to sense DNA damage (*rad9Δ*) reduces the frequency of GLMs, but the cells are unable to respond to the damage and thus die younger. Reducing GLMs by increasing the levels of histone transcription (*hpc2Δ*) or reducing the ability of cells to degrade histones (*tom1Δ*, *ies4Δ*), however, increases the lifespan of cells as it reduces a negative consequence of DDC activation.

A decline in proteostasis, or the ability to ensure proper levels and folding of proteins, is one of the hallmarks of aging. In yeast, activation of the proteasome through increased expression of Rpn4 is sufficient to increase lifespan and enhance proteostasis (*Kruegel et al., 2011*). It is interesting, therefore, that deletion of Tom1, which is involved in proteostatic networks including histone maintenance (*Singh et al., 2009*) and ribosome quality control (*Sung et al., 2016*), is also sufficient

to increase lifespan (*Kruegel et al., 2011*). This suggests a potential distinction between proteostatic processes that are helpful at all ages but decline in function with age (Rpn4/proteasome), and those that introduce tradeoffs between rapid growth in young cells and healthy aging (Tom1).

Single cell analyses like those described here begin to allow for an understanding of heterogeneity within aging populations. While most mother cells we observed experienced one or more GLMs, approximately 25% of wild type cells go their entire lives without a single event. This heterogeneity is under genetic control, as demonstrated by mutations that both increase and decrease GLM frequency and penetrance. Similarly, the likelihood that a GLM will be properly resolved in a given mother cell once it occurs also appears to be variable and under genetic control. Other recent studies have indicated that aging yeast cells can experience different trajectories or 'modes' of aging, as indicated by expression of reporter genes that differ within sub-populations of cells (*Crane and Kaeberlein, 2018*; *Jin et al., 2019*; *Li et al., 2019*). This raises the interesting possibility that not all cells experience the same age-related genome instability, perhaps due to stochastic or inherited factors that influence early life cell fates (*Li et al., 2019*; *Morlot et al., 2019*). Intriguingly, in the Jin, et al. study, they identified two aging paths and cells that died from the second path experienced a reduced mean lifespan, but a similar maximal lifespan to those that died from the first path. This is similar to our situation wherein the prevention of GLMs resulted in an increased mean lifespan, but similar maximal lifespan (*Figure 1—figure supplement 4*). We believe that there is likely to be significant overlap between the cells that follow the second aging path identified by Jin et al, and cells that undergo repeated GLMs. Similarly, GLM events do not occur in a completely stochastic fashion among all cells, as there is history dependence (*Figure 1—figure supplement 4*). This suggests that GLM events could be linked to an underlying cellular state related to the occurrence of DNA damage or reduced repair ability. A powerful feature of microfluidic systems such as the one used here is the ability to resolve, in individual cells, age-associated phenotypes that have previously only been quantified at the population level and to study those phenomena continuously throughout life.

Although it remains unclear whether the specific mechanisms described here are shared during aging in multicellular eukaryotes, genome instability is one of the nine 'Hallmarks of Aging' which are broadly evolutionarily conserved (*López-Otín et al., 2013*). Likewise, the DNA damage repair pathways and checkpoints are generally also broadly conserved in evolution, and defects in these processes are associated with a variety of progeroid syndromes (*Burtner and Kennedy, 2010*; *Lombard et al., 2005*). Aneuploidy and other major chromosomal rearrangements are ubiquitous in age-related human cancers (*Sansregret and Swanton, 2017*), and there is accumulating evidence that rDNA is a key source of replication stress during aging across species (*Flach et al., 2014*; *Pal et al., 2018*). The INO80 subfamily is highly conserved from yeast to humans at both the protein and network topology levels (*Sardiu et al., 2015*), so our findings are likely to have broad relevance to aging. More generally, the antagonistic pleiotropy between rapid and efficient response to DNA damage during youth coming at the expense of age-related declines in function and disease has been proposed in mammalian aging (*Rodier et al., 2007*; *Rufini et al., 2013*). Thus, future studies in this area are likely to help us understand specific mechanisms of cellular and organismal aging and provide insights into age-related pathology which may uncover potential targets for intervention.

## Materials and methods

### Yeast RLS microdissection

Microdissection experiments to determine replicative lifespans were done as previously described (*McCormick et al., 2015*). Briefly, cells were patched onto YPD plates and allowed to grow overnight. Then, cells were arrayed, and virgin daughters were selected for use in the lifespan. New daughters were manually removed from mothers until mother cells die.

### Microfluidics

Cells were imaged using a PDMS microfluidic flow chamber modified from an earlier design (*Crane et al., 2014*) to increase retention over the replicative lifespan of the mother cells. The microfluidic device was composed of multiple chambers in the same fashion as (*Granados et al., 2017*), which allowed individual genotypes to be exposed to identical environments and imaged in the same experiment while being physically isolated. Cells were loaded according to previously

published methods (*Granados et al., 2017*). A volumetric flow rate of 3–7 µL/min per chamber was used, with the flow rate starting low, and increasing during the experiment to improve mother cell retention and to ensure that cells do not aggregate, which can clog the device.

## Microscopy

Cells were imaged using a Nikon TiE-2000 microscope with a 40X oil immersion objective with a 1.3 NA and using the Nikon Perfect Focus System. An enclosed incubation chamber was used to maintain a stable 30˚C environment for the duration of the experiment. Two Aladdin syringe pumps were used for media flow. An LED illumination system (Excelitas 110-LED) was used to provide consistent excitation energies, and to minimize the exposure, illumination was triggered by the camera. Images were acquired using a Hamamatsu Orca Flash 4.0 V2. The microscope was controlled by custom software written in Matlab and Micromanager (https://bitbucket.org/matthew_crane/uscopecontrol; *Crane, 2017*).

Images were corrected for illumination artifacts in two stages. First, to correct for individual differences in the pixel biases, 1000 images were acquired with no illumination, and the individual pixel means were determined which was subtracted from each acquired image. Second, to correct for flatness of field, a fluorescent dye was added to a microfluidic device instead of using a slide with dye. Using a slide containing dye introduces a large amount of out-of-focus light, which results in an underestimation of the field curvature. In order to compensate for the microfluidic features, 1000 images were acquired each with a small offset in the x and y positions. Images were then dilated, and the median value at each location was used. Thus, for each image, the camera bias for that pixel was subtracted, and then it was multiplied by a flatness of field correction factor.

Images were acquired at 5 min intervals for bright-field and fluorescent channels. The fluorescence excitation power was set to 25% for all imaging except the GFP tagged histones, where it was set to 12%. Fluorescence and brightfield light were activated during image acquisition and all other lights in the room were turned off. For bright-field, 3 z-sections were acquired with 2.5 µm intervals, and exposure times of 30 ms were used for automated segmentation and tracking. For the fluorescent channels, 3 z-sections were acquired with 1.5 µm spacing. GFP images were acquired using a Chroma ET49002 filter set, and mCherry images were acquired using a Chroma ET49306. GFP images were acquired using exposure times of 60 ms for all proteins except Htb2 and Hta2 which were acquired using a 30 ms exposure time. mCherry images were acquired using a 60 ms exposure time. These imaging conditions were found to work as a reasonable compromise between the desire for frequent, dense imaging to enable identification of missegregations and retrograde transport, while also minimizing phototoxicity. We performed control experiments to verify that these exposure conditions did not affect the rates of genomic missegregation or replicative lifespan (*Figure 1—figure supplement 2*). Each strain was imaged in multiple independent experimental runs, each with approximately equal numbers of cells.

## Data processing and single cell scoring

Following data acquisition, cells were identified and tracked using previously published software (*Bakker et al., 2018*). This identified the cell outline, and performed initial tracking of the cells through time. To ensure that only young, healthy cells were assessed, we only used cells that were identified in the first three hours of the experiment. Birth events for these cells were then manually scored, and any errors in tracking were corrected. This was all done using the bright-field images. Birth events were scored by multiple observers, and individual cells can be lost from traps prior to death. Data is presented either with or without censoring depending on what would be most appropriate. In the main text, plots aligned based on increasing age used all cells present at that age, even if they were later lost (censored) from the device. For plots aligned by death, only cells that had either died (visibly lost cell wall integrity) or senesced (failed to initiate a new cell division but did not visibly lose cell wall integrity during the experiment) were used. Because censoring in lifespan experiments relies on the assumption that losses are unbiased, we provide replicative lifespan curves both including and excluding censored cells for all strains. Censoring does not change the interpretation or statistical outcome of any of the experiments presented here.

Following annotation of birth events, the fluorescent channel containing the histones was used to observe the GLM dynamics. To ensure consistent scoring across experiments and eliminate bias,

information about the experiments was masked from the scorer until after the data were evaluated. A correction event for a GLM was defined as where the histone fluorescence decreased in the daughter cell while simultaneously increasing in the mother cell. During cell cycles where cells had multiple GLMs during the same cell cycle, only the final event was scored. Events were scored as terminal GLM events if, prior to a correction, the daughter cell visibly separated from the mother cell (indicating cytokinesis) or if the mother died.

## Fluorescence quantification

Quantification of the level of protein localized to the nucleolus (Cdc14) was done using a measure of how asymmetrically distributed the fluorescent signal was. Specifically, we used average brightness of the top 2% of pixels, divided by the cell median. By normalizing to median fluorescence, we corrected for any changes in fluorescence that could occur as a result of photobleaching. This method has been used previously as an accurate measure of the fraction of protein that is nuclear localized (*Cai et al., 2008*; *Granados et al., 2017*).

## Yeast strains and growth

The GFP strains were all acquired from the yeast GFP collection (*Huh et al., 2003*). The Htb2: mCherry strain was created by mating and sporulation of the strain from *Granados et al. (2017)*. This strain was then crossed with the relevant GFP strains (Nup49:GFP, Myo1:GFP, Tub1:GFP, Spc72:GFP, Cdc14:GFP) or deletion strains (*hpc2Δ*, *fob1Δ*, *spt21Δ*, *tom1Δ*, *mad3Δ*) from the deletion collection (*Winzeler et al., 1999*) and then confirmed by colony PCR. The LacI-GFP strains with 50 LacO repeats on ChrXII was obtained from *Ide et al. (2010)*. The strains containing TetR-GFP and TetO repeats integrated into ChrIV or ChrV were obtained from *Fernius and Marston (2009)*. These were then crossed with the strain containing Htb2:mCherry. Complete list of strains available in *Supplementary file 1*.

Prior to each microfluidic experiment, single colonies were picked into SC media (Sunrise Biosciences) with 2% dextrose. Cells were grown overnight, and then diluted 1:200 in fresh media and grown for 5–6 hr. Prior to loading into the microfluidic device, 0.5 mL of SC 2% dextrose with 0.5% BSA was added to each 5 mL culture to prevent the cells from adhering to the PDMS during loading. During experiments, SC media with 2% dextrose and 0.1% BSA was used, and cells were imaged for 72 hr.

## Statistical analysis

Error bars in the figures which contained bar plots were generated by bootstrapping with replacement, and then determining the 95% confidence intervals. Error bars in figures with line plots are standard error. Statistical significance for lifespan was determined using the log-rank test. Log-rank test was performed with, and without, censored cells that were lost prior to senescence or death. To compare distributions (such as numbers of GLM events over the lifespan), a two-tailed t-test assuming equal variance was used. Correlations between GLM events and remaining replicative lifespan were calculated with the Spearman correlation using the population of cells alive at each replicative age. Cochran's q-test was used to determine whether there was an age-related increase in GLM rates for individual strains.

## Differences between censored and uncensored survival data

Frequently in experiments or clinical studies that involve the generation of survival curves, some samples will be removed from the population under observation. For example, a patient may leave a study not because of death, but because they move to a different country. This can be treated in a relatively straightforward manner statistically by including these individuals in the analysis until the point that they are lost (or censored). This relies on the assumption that there is no bias in whether a sample is lost or retained. A recurring concern with microfluidic aging experiments involving yeast is whether there is a bias in how cells are lost or retained. This appears especially important when the mutation or transgene affects cell morphology or cell cycle, as this can result in a bias in which cells are lost from the traps. To reduce the likelihood that our observations were directly affected by loss rates, in the main text we have plotted all cells that were present at that replicative age for plots from birth. Thus, if a cell was lost at replicative age 20, it was included in the plots until age 19. For

all plots that are aligned by death, only cells that die in the device are used. Given that this is an altered population distribution and smaller number of cells, these plots are slightly different, but they do not affect the conclusions. For bar charts showing the fraction of cells that die from a terminal missegregation, only cells that die in the device are included. For the survival curves in the main text that show individual cells and cell cycles that had a GLM (*Figure 1H* for example), only cells that die in the device are included. For replicative lifespans shown in the supplementary, we include survival curves with and without censored cells.

## Aligning cells from birth or from death

Cells can be aligned either by birth (counting up from replicative age = 0), or by death (counting back from death). Either processing makes some assumptions about how similar cells are to one another. If cells are most similar to each other when they are born, aligning by birth makes sense, and as the replicative age increases, the number of samples decreases because cells are removed by death or senescence. In contrast, assuming that cells are similar at death implies that the phenotype of interest is most similar as cells approach death. For example, the average time cells take to proceed through each cell cycle increases geometrically when cells are aligned by birth, but exponentially when aligned by death. To demonstrate the increase in GLM rates as individual cells approach death, we align the cells by death in *Figure 1*. To show the increase in GLM rates during aging, cells are aligned by birth for the rest of the figures.

## Effect of fluorophore and histone tagged to visualize chromatin

In order to determine the effect of tagging different histone proteins, we compared the effect of tagging different histone proteins, and compared the effects of using two different fluorophores (GFP and mCherry). Both of Hta2:GFP and Htb2:mCherry strains were found to have similar numbers of missegregation events during their lifetimes, and similar fractions of these events were corrected (*Figure 1—figure supplement 1*). Similarly, both the strain containing Hta2:GFP, and that containing Htb2:mCherry experience a similar increase in GLMs as they approach the end of their lives. The most notable distinction between the strains are the replicative lifespans, with Hta2:GFP experiencing what we consider to be a normal lifespan for the BY background (*Supplementary file 2*, *Figure 1—figure supplement 1*). The strain with Htb2:mCherry, however, had a somewhat shorter lifespan (*Supplementary file 2*, *Figure 1—figure supplement 1*). Removing FOB1, however, results in an increase of the replicative lifespan of the Htb2:mCherry strain by ~30%, which is in line with results from literature (*McCormick et al., 2015*). Furthermore, the increase in replicative lifespan as a result of increased histone transcription has been less thoroughly studied, but our results are in line with those previously reported by another group (*Feser et al., 2010*; *Kruegel et al., 2011*). In order to determine whether the primary cause of the lifespan reduction in the Htb2:mCherry strain was the histone selected, or the fluorescent reporter, we also obtained lifespans for Htb2:GFP (*Supplementary file 2*, *Figure 1—figure supplement 1*). Tagging Htb2 with GFP results in a lifespan that is statistically indistinguishable from the Hta2:GFP strain (p>0.05 logrank test). Thus, although there is an unexpected reduction in lifespan for the Htb2:mCherry strain, we do not believe that it affects our results.

Likewise, we determined the correlation between missegregation events and remaining lifespan at the single cell level. The correlation is between the binary presence or absence of a missegregation event at a specific age, and the remaining lifespan. Strikingly, as shown in *Figure 1—figure supplement 4*, for both strains, the correlation between missegregation events and remaining replicative lifespan is the same for both Htb2:mCherry and Hta2:GFP. This is in spite of the difference in absolute lifespan between the two strains.

Because GFP fluorescence is much more affected than mCherry fluorescence by changing pH (*Shaner et al., 2005*), and the pH of the cytoplasm in aging yeast has previously been shown to increase (*Henderson et al., 2014*), we chose to perform the majority of the experiments using mCherry. This ensured that any changes in pH homeostasis during aging would not affect our histone imaging.

## Acknowledgements
We would particularly like to thank S Biggins, B Brewer and M Raghuraman for constructive discussions. We also thank L Veenhoff and Kaeberlein lab members for feedback and advice. Strains YSI129, YSI130, AMY914 and AMY1081 were generous gifts from Jessica Tyler and Adele Marston. This work was supported by NIH grants T32AG000057, R01AG056359, and P30AG013280.

## Additional information

### Competing interests
Matt Kaeberlein: Reviewing editor, *eLife*. The other authors declare that no competing interests exist.

### Funding

| Funder | Grant reference number | Author |
|---|---|---|
| National Institute on Aging | T32AG000057 | Matt Crane |
| National Institute on Aging | R01AG056359 | Matt Kaeberlein |
| National Institute on Aging | P30AG013280 | Matt Kaeberlein |

The funders had no role in study design, data collection and interpretation, or the decision to submit the work for publication.

### Author contributions
Matthew M Crane, Conceptualization, Formal analysis, Investigation, Visualization, Methodology, Writing—original draft, Writing—review and editing; Adam E Russell, Brent J Schafer, Riley Whalen, Jared Almazan, Mung Gi Hong, Bao Nguyen, Joslyn E Goings, Kenneth L Chen, Ryan Kelly, Investigation, Methodology; Ben W Blue, Investigation, Visualization, Methodology; Matt Kaeberlein, Conceptualization, Supervision, Funding acquisition, Writing—original draft, Writing—review and editing

### Author ORCIDs
Matthew M Crane (iD) https://orcid.org/0000-0002-6234-0954
Matt Kaeberlein (iD) https://orcid.org/0000-0002-1311-3421

### Decision letter and Author response
Decision letter https://doi.org/10.7554/eLife.50778.031
Author response https://doi.org/10.7554/eLife.50778.032

## Additional files
### Supplementary files
• Supplementary file 1. List of strains and genotypes used in this study.
DOI: https://doi.org/10.7554/eLife.50778.025
• Supplementary file 2. List of replicative lifespans for strains used.
DOI: https://doi.org/10.7554/eLife.50778.026
• Transparent reporting form DOI: https://doi.org/10.7554/eLife.50778.027

### Data availability
Data are available on Dryad at https://doi.org/10.5061/dryad.cz8w9ghzx.

The following dataset was generated:

| Author(s) | Year | Dataset title | Dataset URL | Database and Identifier |
|---|---|---|---|---|
| Crane MM, Russell | 2019 | Data from: DNA damage | https://doi.org/10.5061/ | Dryad Digital |

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
