## [Decision Letter]

Thank you for submitting your article "DNA damage checkpoint activation impairs chromatin homeostasis and promotes mitotic catastrophe during aging" for consideration by *eLife*. Your article has been reviewed by three peer reviewers, and the evaluation has been overseen by Nan Hao as the Reviewing Editor and Jessica Tyler as the Senior Editor. Reviewer #3 was Scott McIsaac, and the other reviewers have opted to remain anonymous.

The reviewers have discussed the reviews with one another and the Reviewing Editor has drafted this decision to help you prepare a revised submission.

Summary:

In this manuscript, Crane et al. use a microfluidic device they developed previously, to track individual budding yeast cells through their entire replicative lifespan. This is not the first study to do so, but it is the first one to carefully monitor the segregation of the genetic material from generation to generation. They make an interesting finding that aging cells undergo occasional chromatin missegregation events during cell cycle (defined as "GLMs"). They further show that the frequency of missegregation increases during aging and, by combining the histone reporter with other reporters and knockouts, they claim that age-associated DNA damage and activation of the DNA damage checkpoint (DDC) leads to a drop in histone levels, which in turn triggers missegregation of the entire nucleus. The reviewers think this paper is a significant advance for studying cellular aging and cell division, but also raise a number of concerns that must be adequately addressed before the paper can be accepted.

Essential revisions:

1) It has been long observed that almost every aging cell undergoes late-life cell cycle extension. It seems that GLMs, which features abnormal segregation of whole chromosomes, only occur in a fraction of cells and can only account for a part of age-dependent cell cycle extension events. Related to that, while their data support a role for DNA damage, we are not sure that DNA damage may be the only way to get to GLMs. As they point out, GLMs were also reported in aging cells by Neurohr et al. But those authors invoked dysregulation of the G1/S transition in their model (i.e., much earlier in the cell cycle than the metaphase DDC). Likewise, some of the phenotypes the authors see (e.g., both SPBs moving in the daughter cell) have also been seen in nuclear positioning mutants, which starts in G1 and also involves cortical attachment processes, well before the metaphase DNA damage and spindle positioning checkpoints are engaged. Furthermore, in Figure 5, the same life-long reduction in GLMs is observed in *ies4*∆ and *tom1*∆ cells. It seems that there are multiple routes increasing GLMs, which may or may not depend on DDC. Finally, a recent Cell Reports paper argues that 'Cell-cycle regulation is involved in the age-associated DNA repair inefficiency' and points also to G1 length as a key determinant (Young et al., 2019). From their existing datasets, it may be informative to break down the duration of individual cell cycle phases and also measure nuclear positioning from G1 onwards, as the mother cells progress in the cell cycle. We request data re-analysis of cell cycle kinetics and careful discussions to clarify the relationship among DDC, GLMs and age-dependent cell cycle extension in general.

2) The authors claim that the activation of DDC causes GLMs. Could DDC activation in young cells also lead to GLMs, or some age-related factors are also required? We would request an experiment to induce DDC in young cells using chemical or radiation agents and observe whether GLMs will occur. This will help to establish the causality between DDC and GLMs, and to determine whether GLMs are a consequence of DDC, aging, or both.

3) We understand that the appearance and resolution of GLMs could be a determining factor of lifespan for some cells, but some of the data suggest that GLMs might not be a major driver of replicative potential in many cases. For example, the fraction of mothers with GLMs in two control experiments are identical (Figure 1—figure supplement 1B), yet the median/maximal RLSs appear quite different (Figure 1—figure supplements 1D and E). In addition, *fob1*∆ cells, which live longer than wt cells (on average), have slightly less GLMs than wt, but *rad9*∆ cells, which have almost no GLMs, are substantially shorter-lived than wt. Related to these, in Figure 1—figure supplement 4, the effect of GLM resolution is only evident for the median lifespan, and not at all for the maximal lifespan. It seems that whether the cells correct GLMs or not has no effect on the maximal lifespan. Typically, for interventions that strongly and significantly extend longevity (many of which discovered over the years by the authors), it is both the maximal and the median lifespan that are extended. Why isn't this the case here? We recommend that the authors include a discussion to clarify the conditions where GLMs may or may not be a major driver of RLS potential and modify/improve the model schematic in Figure 7 that is a little unclear.

---

## [Author Response]

Essential revisions:1) It has been long observed that almost every aging cell undergoes late-life cell cycle extension. It seems that GLMs, which features abnormal segregation of whole chromosomes, only occur in a fraction of cells and can only account for a part of age-dependent cell cycle extension events. Related to that, while their data support a role for DNA damage, we are not sure that DNA damage may be the only way to get to GLMs. As they point out, GLMs were also reported in aging cells by Neurohr et al. But those authors invoked dysregulation of the G1/S transition in their model (i.e., much earlier in the cell cycle than the metaphase DDC). Likewise, some of the phenotypes the authors see (e.g., both SPBs moving in the daughter cell) have also been seen in nuclear positioning mutants, which starts in G1 and also involves cortical attachment processes, well before the metaphase DNA damage and spindle positioning checkpoints are engaged. Furthermore, in Figure 5, the same life-long reduction in GLMs is observed in ies4∆ and tom1∆ cells. It seems that there are multiple routes increasing GLMs, which may or may not depend on DDC. Finally, a recent Cell Reports paper argues that 'Cell-cycle regulation is involved in the age-associated DNA repair inefficiency' and points also to G1 length as a key determinant (Young et al., 2019). From their existing datasets, it may be informative to break down the duration of individual cell cycle phases and also measure nuclear positioning from G1 onwards, as the mother cells progress in the cell cycle. We request data re-analysis of cell cycle kinetics and careful discussions to clarify the relationship among DDC, GLMs and age-dependent cell cycle extension in general.

We agree with the reviewer, and have clarified the text on these points. The activation of the metaphase DDC is the dominant cause of the age-related increase in GLM events, but it is not necessarily the sole cause. When the DDC is compromised by removal of Rad9, the number of GLMs that occurs during aging is reduced by 95%, but not completely eliminated. Compromising the spindle assembly checkpoint (Figure 3—figure supplement 2), or spindle positioning checkpoint (Figure 7) failed to affect the age-related increase in GLM rates – which further suggests that activation of the DDC is the dominant cause of age-related GLMs. We believe that the reduction in GLM rates in both the *ies4*∆ and *tom1*∆ cells is a result of their roles in the DDC, and thus in the same dominant pathway.

We agree that changes to the G1 duration during aging could be critical to the cells ability to respond to DNA damage and the frequency of GLMs. To respond to this question raised by the reviewers, we have performed additional experiments and data analysis. Using a strain with WHI5:GFP (a yeast analog of Rb that helps determine the duration of G1) and Htb2:mCherry, we imaged cells during normal aging the microfluidic device, and then annotated birth events and GLMs. Using this data, and the Whi5 traces, we determined how the G1 duration changed during aging in wild-type cells, and whether cells that experienced GLMs had different G1 durations than those that did not. Given the link between extended G1 duration and reduced single strand annealing rates, we expected to find cell cycles that underwent GLMs longer G1 durations. As expected, we confirmed prior reports showing that the time spent in G1 increases in individual cells both in absolute terms, and as a fraction of the of the cell cycle. Importantly, however, we found no difference in the G1 duration between cell cycles where a cell underwent a GLM and cell cycles which proceeded normally.

This new data and analysis is included in Figure 1—figure supplement 5. A potential explanation is that although extended G1 durations reduce single strand annealing rates, that could be compensated for by increased MCM loading, and improved origin firing. This, or other beneficial aspects of extended G1 durations, may compensate and explain why we fail to identify a connection between G1 duration and GLM frequency. We include additional commentary on this in the revised Discussion. The relationship between cell cycle duration and age will continue to be an exciting area of research, but further work is beyond the scope of this paper.

2) The authors claim that the activation of DDC causes GLMs. Could DDC activation in young cells also lead to GLMs, or some age-related factors are also required? We would request an experiment to induce DDC in young cells using chemical or radiation agents and observe whether GLMs will occur. This will help to establish the causality between DDC and GLMs, and to determine whether GLMs are a consequence of DDC, aging, or both.

We appreciate this insightful suggestion and have conducted an experiment in young cells to test this. We placed cells expressing Htb2:mCherry in the microfluidic device, and allowed them to grow and acclimate for 3 hours. Then the media was changed and cells were exposed to 500 µg/ml zeocin for 7 hours. Zeocin intercalates into DNA and causes double stranded breaks, and was used by the Gasser lab in their studies of histone degradation following DNA damage. We used dose of 500 µg/ml zeocin based on their work. During the 7 hours of exposure, cells were imaged every 5 minutes, and following the experiment GLMs were scored. The continuous exposure to zeocin resulted in a high rate of GLMs, with the average time to GLM being three hours following initial exposure. The number of GLMs was dramatically higher in the zeocin treated cells compared with the control cells.

We have included this new experiment and data in the revised version of the paper in Figure 3—figure supplement 1. This experiment reinforces the causal link between DNA damage and GLMs.

3) We understand that the appearance and resolution of GLMs could be a determining factor of lifespan for some cells, but some of the data suggest that GLMs might not be a major driver of replicative potential in many cases. For example, the fraction of mothers with GLMs in two control experiments are identical (Figure 1—figure supplement 1B), yet the median/maximal RLSs appear quite different (Figure 1—figure supplements 1D and E). In addition, fob1∆ cells, which live longer than wt cells (on average), have slightly less GLMs than wt, but rad9∆ cells, which have almost no GLMs, are substantially shorter-lived than wt. Related to these, in Figure 1—figure supplement 4, the effect of GLM resolution is only evident for the median lifespan, and not at all for the maximal lifespan. It seems that whether the cells correct GLMs or not has no effect on the maximal lifespan. Typically, for interventions that strongly and significantly extend longevity (many of which discovered over the years by the authors), it is both the maximal and the median lifespan that are extended. Why isn't this the case here? We recommend that the authors include a discussion to clarify the conditions where GLMs may or may not be a major driver of RLS potential and modify/improve the model schematic in Figure 7 that is a little unclear.

We appreciate the insight into the number and frequency of GLMs within a population, and have provided additional clarification in both the Discussion and the Results on these points. We address each of the points in detail below.

With regard to the *control experiments*, and the frequency of GLMs. Comparing the Hta2:GFP, Htb2:GFP and Htb2:mCherry tagged strains, there is a difference in lifespan, in that the GFP tagged strains live consistently longer than the mCherry tagged strain; however, the total fraction of cells that experience GLMs, appears equal between the strains (Figure 1—figure supplement 1B). On average, however, the total number of GLMs experienced by the shorter lived mCherry strain is higher (Figure 1—figure supplement 1C). The GLMs do occur in close proximity to the end of life in both GFP and mCherry strains (Figure 1—figure supplement 1A).

With regards to the *reduced GLM rate in rad9∆ cells*. The extended activation of the DDC can be deleterious and result in GLMs and terminal missegregations wherein the mother cell loses its genome and terminally senesce. Activation of the DDC when DNA damage has occurred, however, is necessary for a full lifespan. A simple analogy to DDC activation would be hospital visits. We want to decrease hospital visits (extended DDC activation that results in GLMs) by improving care, not by removing hospitals (*rad9*∆ cells). In the case of *fob1*∆, *hpc2*∆, *tom1*∆ and *ies4*∆ cells, the DDC is still capable of activating if/when necessary – it simply isn’t required to activate for extended periods of time. The reduced ability of the DDC to activate in the presence of DNA damage in *rad9*∆ cells results in an increased mortality rate.

With regards to the *increased median lifespan, but unchanged maximal lifespan* in Figure 1—figure supplement 4. This is an extremely interesting point the reviewers raise, and one which we believe deserves additional scrutiny in the future. Recently several labs have demonstrated population heterogeneity in aging trajectories at the single cell level in yeast. With a single isogenic population, the individual cells may die from different physiological failures. In the recently published Jin et al., 2019 paper cells that died from the second aging trajectory had a reduced mean lifespan, but identical maximal lifespan. Thus, similar to our analysis of GLMs, shifting cells from aging path 2 to aging path 1 in Jin et al. would increase the mean lifespan, but not change the maximal lifespan. We believe that there is going to be significant overlap between the cells that experience GLMs, and the aging trajectories identified by Jin et al. We have modified the Discussion to include commentary on this likely relationship.